# THE EXPRESSIVE LEAKY MEMORY NEURON: AN EFFICIENT AND EXPRESSIVE PHENOMENOLOGICAL NEURON MODEL CAN SOLVE LONG-HORIZON TASKS

Aaron Spieler[1,2], Nasim Rahaman[3,2], Georg Martius[1,2], Bernhard Schölkopf[2], and Anna Levina[1,4]

[1]University of Tübingen, Germany
[2]Max Planck Institute for Intelligent Systems, Tübingen, Germany
[3]Mila, Quebec AI Institute, Canada
[4]Max Planck Institute for Biological Cybernetics, Tübingen, Germany

## ABSTRACT

Biological cortical neurons are remarkably sophisticated computational devices, temporally integrating their vast synaptic input over an intricate dendritic tree, subject to complex, nonlinearly interacting internal biological processes. A recent study proposed to characterize this complexity by fitting accurate surrogate models to replicate the input-output relationship of a detailed biophysical cortical pyramidal neuron model and discovered it needed temporal convolutional networks (TCN) with millions of parameters. Requiring these many parameters, however, could stem from a misalignment between the inductive biases of the TCN and cortical neuron's computations. In light of this, and to explore the computational implications of leaky memory units and nonlinear dendritic processing, we introduce the Expressive Leaky Memory (ELM) neuron model, a biologically inspired phenomenological model of a cortical neuron. Remarkably, by exploiting such slowly decaying memory-like hidden states and two-layered nonlinear integration of synaptic input, our ELM neuron can accurately match the aforementioned input-output relationship with under ten thousand trainable parameters. To further assess the computational ramifications of our neuron design, we evaluate it on various tasks with demanding temporal structures, including the Long Range Arena (LRA) datasets, as well as a novel neuromorphic dataset based on the Spiking Heidelberg Digits dataset (SHD-Adding). Leveraging a larger number of memory units with sufficiently long timescales, and correspondingly sophisticated synaptic integration, the ELM neuron displays substantial long-range processing capabilities, reliably outperforming the classic Transformer or Chrono-LSTM architectures on LRA, and even solving the Pathfinder-X task with over 70% accuracy (16k context length). These findings raise further questions about the computational sophistication of individual cortical neurons and their role in extracting complex long-range temporal dependencies.

## 1 INTRODUCTION

The human brain has impressive computational capabilities, yet the precise mechanisms underpinning them remain largely undetermined. Two complementary directions are pursued in search of mechanisms for brain computations. On the one hand, many researchers investigate how these capabilities could arise from the collective activity of neurons connected into a complex network structure Maass (1997); Gerstner & Kistler (2002); Grüning & Bohte (2014), where individual neurons might be as basic as leaky integrators or ReLU neurons. On the other hand, it has been proposed that the intrinsic computational power possessed by individual neurons Koch (1997); Koch & Segev (2000); Silver (2010) contributes a significant part to the computations.

Even though most work focuses on the former hypothesis, an increasing amount of evidence indicates that cortical neurons are remarkably sophisticated Silver (2010); Gidon et al. (2020); Larkum (2022), even comparable to expressive multilayered artificial neural networks Poirazi et al. (2003); Jadi et al. (2014); Beniaguev et al. (2021); Jones & Kording (2021), and capable of discriminating between dozens to hundreds of input patterns Gütig & Sompolinsky (2006); Hawkins & Ahmad (2016);

Moldwin & Segev (2020). Numerous biological mechanisms, such as complex ion channel dynamics (e.g. NMDA nonlinearity Major et al. (2013); Lafourcade et al. (2022); Tang et al. (2023)), plasticity on various and especially longer timescales (e.g. slow spike frequency adaptation Kobayashi et al. (2009); Bellec et al. (2018)), the intricate cell morphology (e.g. nonlinear integration by dendritic tree Stuart & Spruston (2015); Poirazi & Papoutsi (2020); Larkum (2022)), and their interactions, have been identified to contribute to their complexity.

Detailed biophysical models of cortical neurons aim to capture this inherent complexity through high-fidelity mechanistic simulations Hay et al. (2011); Herz et al. (2006); Almog & Korngreen (2016). However, they require a lot of computing resources to run and typically operate at a very fine level of granularity that does not facilitate the extraction of higher-level insights into the neuron's computational principles. A promising approach to derive such higher-level insights from simulations is through the training of surrogate phenomenological neuron models. Such models are designed to replicate the output of biophysical simulations but use simplified interpretable components. This approach was employed, for example, to model computation in the dendritic tree via simple two-layer ANN Poirazi et al. (2003); Tzilivaki et al. (2019); Ujfalussy et al. (2018). Building on this line of research, a recent study by Beniaguev et al. (2021) developed a temporal convolutional network to capture the spike-level input/output (I/O) relationship with millisecond precision, accounting for the complexity of integrating diverse synaptic input across the entirety of the dendritic tree of a high-fidelity biophysical neuron model. It was found that a highly expressive temporal convolutional network with millions of parameters was essential to reproduce the aforementioned I/O relationship.

In this work, we propose that a model equipped with appropriate biologically inspired components that align with the high-level computational principles of a cortical neuron should be capable of capturing the I/O relationship using a substantially smaller model size. To achieve this, a model would likely need to account for multiple mechanisms of neural expressivity and judiciously allocate computational resources and parameters in a rough analogy to biological neurons. Should such a construction be possible, the required design choices may yield insights into principles of neural computation at the conceptual level. We proceed to design the Expressive Leaky Memory (ELM) neuron model (see Figure 1), a biologically inspired phenomenological model of a cortical neuron. While biologically inspired, low-level biological processes are abstracted away for computational efficiency, and consequently, individual parameters of the ELM neuron are not designed for direct biophysical interpretability. Nevertheless, model ablations can provide conceptual insights into the computational components required to emulate the cortical input/output relationship. The ELM neuron functions as a recurrent cell and can be conveniently used as a drop-in replacement for LSTMs Hochreiter & Schmidhuber (1997).

Our experiments show that a variant of the ELM neuron is expressive enough to accurately match the spike level I/O of a detailed biophysical model of a layer 5 pyramidal neuron at a millisecond temporal resolution with a few thousand parameters, in stark contrast to the millions of parameters required by temporal convolutional networks. Conceptually, we find accurate surrogate models to require multiple memory-like hidden states with longer timescales and highly nonlinear synaptic integration. To explore the implications of neuron-internal timescales and sophisticated synaptic integration into multiple memory units, we first probe its temporal information integration capabilities on a challenging biologically inspired neuromorphic dataset requiring the addition of spike-encoded spoken digits. We find that the ELM neuron can outperform classic LSTMs leveraging a sufficient number of slowly decaying memory and highly nonlinear synaptic integration. We subsequently evaluate the ELM neuron on the well-established long sequence modeling LRA benchmarks from the machine learning literature, including the notoriously challenging Pathfinder-X task, where it achieves over $70\%$ accuracy but many transformer-based models do not learn at all.

**Our contributions are the following.**

1. We propose the phenomenological Expressive Leaky Memory (ELM) neuron model, a recurrent cell architecture inspired by biological cortical neurons.

2. The ELM neuron efficiently learns the input/output relationship of a sophisticated biophysical model of a cortical neuron, indicating its inductive biases to be well aligned.

3. The ELM neuron facilitates the formulation and validation of hypotheses regarding the underlying high-level neuronal computations using suitable architectural ablations.

4. Lastly, we demonstrate the considerable long-sequence processing capabilities of the ELM neuron through the use of long memory and synapse timescales.

## 2   THE EXPRESSIVE LEAKY MEMORY NEURON

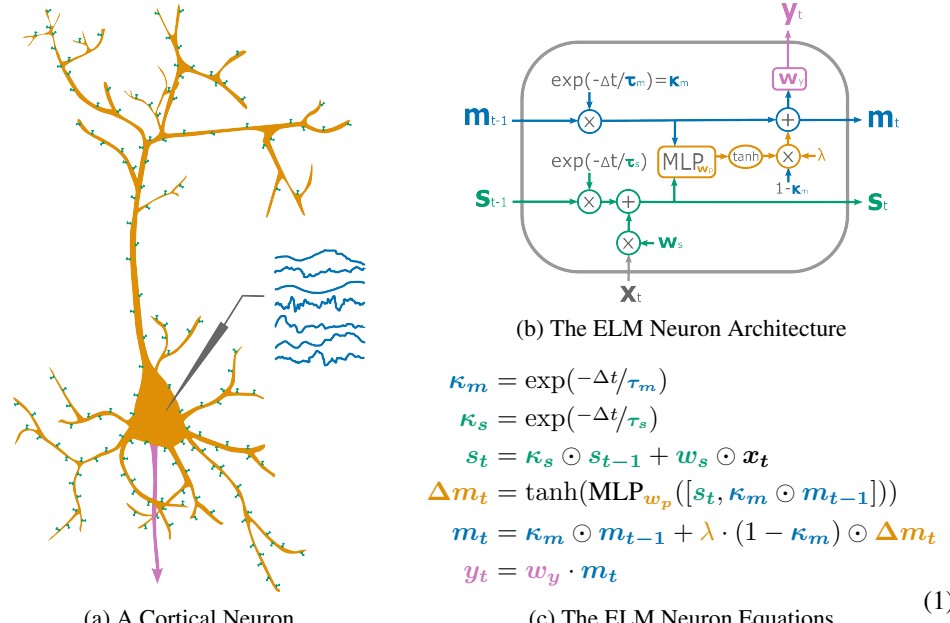

(b) The ELM Neuron Architecture

$$
\begin{aligned}
\kappa_m &= \exp(-\Delta t/\tau_m) \\
\kappa_s &= \exp(-\Delta t/\tau_s) \\
s_t &= \kappa_s \odot s_{t-1} + w_s \odot x_t \\
\Delta m_t &= \tanh(\mathrm{MLP}_{w_p}([s_t, \kappa_m \odot m_{t-1}])) \\
m_t &= \kappa_m \odot m_{t-1} + \lambda \cdot (1 - \kappa_m) \odot \Delta m_t \\
y_t &= w_y \cdot m_t
\end{aligned}
\tag{1}
$$

(a) A Cortical Neuron          (c) The ELM Neuron Equations

Figure 1: **The biologically motivated Expressive Leaky Memory (ELM) neuron model**. The architecture can be divided into the following components: the input current synapse dynamics, the integration mechanism dynamics, the leaky memory dynamics, and the output dynamics. **a)** Sketch of a biological cortical pyramidal neuron segmented into the analogous architectural components using the corresponding colors. **b)** Schematics of the ELM neuron architecture, component-wise colored accordingly. **c)** The ELM neuron equations, where $x_t \in \mathbb{R}^{d_s}$ is the input at time $t$, $\Delta t \in \mathbb{R}^+$ the fictitious elapsed time in milliseconds between two consecutive inputs $x_{t-1}$ and $x_t$, $m \in \mathbb{R}^{d_m}$ are memory units, $s \in \mathbb{R}^{d_s}$ the synapse currents (traces), $\tau_m \in \mathbb{R}^{+^{d_m}}$ and $\tau_s \in \mathbb{R}^{+^{d_s}}$ their respective timescales in milliseconds, $w_s \in \mathbb{R}^{+^{d_s}}$ are synapse weights, $w_p$ the weights of a *Multilayer Perceptron* (MLP) with $l_{\mathrm{mlp}}$ hidden layers of size $d_{\mathrm{mlp}}$, $w_y \in \mathbb{R}^{d_o \times d_m}$ the output weights, $\lambda \in \mathbb{R}^+$ a scaling factor for the delta memory $\Delta m_t \in \mathbb{R}^{d_m}$, and $y \in \mathbb{R}^{d_o}$ the output.

In this section, we discuss the design of the Expressive Leaky Memory (ELM) neuron, and its variant Branch-ELM. Its architecture is engineered to capture sophisticated cortical neuron computations efficiently. Abstracting mechanistic neuronal implementation details away, we resort to an overall recurrent cell architecture with biologically motivated computational components. This design approach emphasizes conceptual over mechanistic insight into cortical neuron computations.

**The current synapse dynamics.**   Neurons receive inputs at their synapses in the form of sparse binary events known as spikes Kandel et al. (2000). While the Excitatory/Inhibitory synapse identity determines the sign of the input (always given), the positive synapse weights can act as simple input gating $w_s$ (learned in Branch-ELM). The synaptic trace $s_t$ denotes a filtered version of the input, believed to aid coincidence detection and synaptic information integration in neurons König et al. (1996). This implementation is known as the current-based synapse dynamic Dayan & Abbott (2005).

**The memory unit dynamics.**   The state of a biological neuron may be characterized by diverse measurable quantities, such as their membrane voltage or various ion/molecule concentrations (e.g. Ca+, mRNA, etc.), and their rate of decay over time (slow decay <-> large timescale), endowing them with a sort of **leaky memory** Kandel et al. (2000); Dayan & Abbott (2005). However, which of these quantities are computationally relevant, how and where they interact, and on what timescale, remains a topic of active debate Aru et al. (2020); Herz et al. (2006); Almog & Korngreen (2016); Koch (1997); Chavlis & Poirazi (2021); Cavanagh et al. (2020); Gjorgjieva et al. (2016). Therefore, to match a biological neuron's computations, the surrogate model architecture needs to be **expressive**

enough to accommodate a large range of possibilities. In the ELM neuron, we achieve this by making the number of memory units $d_m$ a hyper-parameter and equipping each of them with a $\tau_m$ (always learnable), setting it apart most other computational neuroscience models.

**The integration mechanism dynamics.** This dynamic refers to how the synaptic input $s_t$ is integrated into the memory units $\Delta m_t$ in analogy to the dendritic tree of a cortical neuron. While earlier perspectives suggested an integration process akin to linear summation Jolivet et al. (2008), newer studies advocate for complex nonlinear integration Almog & Korngreen (2016); Gidon et al. (2020); Larkum (2022), specifically proposing multi-layered ANNs as more suitable descriptions Poirazi et al. (2003); Jadi et al. (2014); Marino (2021); Jones & Kording (2021); Iatropoulos et al. (2022); Jones & Kording (2022); Hodassman et al. (2022), also backed by recent evidence of neuronal plasticity beyond synapses Losonczy et al. (2008); Holtmaat et al. (2009); Abraham et al. (2019). Motivated by this ongoing discussion, we choose to parameterize the input integration using a Multilayer Perceptron (MLP) ($w_p$ always learnable, with $l_{\text{mlp}} = 1$ and $d_{\text{mlp}} = 2d_m$), which can be used to explore the full range of hypothesized integration complexities, while offering a straightforward way to quantify and ablate the ELM neuron integration complexity. In the Branch-ELM variant (for motivation see details in Section 4 and Figure 4) we extend the integration mechanism dynamics; before the MLP is applied, the synaptic input $s_t \in \mathbb{R}^{d_s}$ is reduced to a smaller number ($d_{\text{tree}}$) of branch activations, each computed as a sum over $d_{\text{brch}}$ neighboring synaptic inputs (with $d_{\text{tree}} * d_{\text{brch}} = d_s$). In this variant the $w_s$ need to be learnable, as they are responsible for weighting the sum and cannot be absorbed in the MLP later. Despite the biological inspiration, the MLP and synapses are only intended to capture the **neuron analogous plasticity and dendritic nonlinearity**, and cannot give a mechanistic explanation of these phenomena in neuron. Finally, incorporating previous memory units $m_{t-1}$ into the integration process, the ELM can accommodate state-dependent synaptic integration and related computations Hodgkin & Huxley (1952); Gasparini & Magee (2006); Bicknell & Häusser (2021), and enables the relationships among memory units $m$ to be fully learnable. The range of the $m$ values is controlled by $\lambda$, and the mixing of the proposal values by the parameter $k_m$ (for details on parameters see Appendix Section A). Crucially, our approach sidesteps the need for expert-designed and pre-determined differential equations typical in phenomenological neuron modeling.

**The output dynamics.** Spiking neurons emit their output spike at the axon hillock roughly when their membrane voltage crosses a threshold Kandel et al. (2000). The ELM neuron's output is similarly based on its internal state $m_{t-1}$ (using a linear readout layer $w_y$), which rectified can be interpreted as the spike probability. For task compatibility, the output dimensionality is adjusted based on the respective dataset (not affecting neuron expressivity).

## 3 RELATED WORK

**Accurately replicating the full spike-level neuron input/output (I/O)** relationship of detailed biophysical neuron models at millisecond resolution in a computationally efficient manner presents a formidable challenge. However, addressing this dynamics-learning task could yield valuable insights into neural mechanisms of expressivity, learning, and memory Durstewitz et al. (2023). The relative scarcity of prior work on this subject can be partially attributed to the computational complexity of cortical neurons only recently garnering increased attention Tzilivaki et al. (2019); Beniaguev et al. (2021); Larkum (2022); Poirazi & Papoutsi (2020). Additionally, traditional phenomenological neuron models have primarily aimed to replicate specific computational phenomena of neurons or networks Koch (1997); Izhikevich (2004); Herz et al. (2006), rather than the entire I/O relationship.

**Phenomenological neuron modeling** research on temporally or spatially less detailed I/O relationship of biophysical neurons has been primarily centered around the use of multi-layered ANN structures in analogy to the neurons dendritic tree Poirazi et al. (2003); Tzilivaki et al. (2019); Ujfalussy et al. (2018). Similarly, we parametrize the synaptic integration with an MLP, while crucially extending this modeling perspective in several ways. Drawing upon the principles of classical phenomenological modeling via differential equations Izhikevich (2004); Dayan & Abbott (2005), our approach embraces the recurrent nature of neurons. We further consider the significance of hidden states $m$ beyond membrane voltage, as seen in prior works with predetermined variables Brette & Gerstner (2005); Gerstner et al. (2014). This addition enables us to flexibly investigate internal memory timescales $\tau_m$.

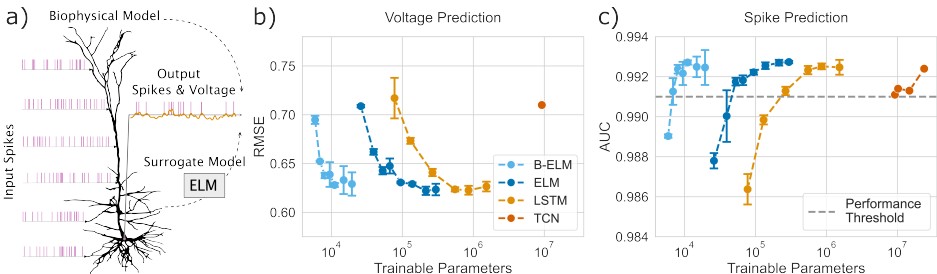

Figure 2: **The ELM neuron is a computationally efficient model of cortical neuron. a)** detailed biophysical model of a layer 5 cortical pyramidal cell was used to generate the NeuronIO dataset consisting of input spikes and output spikes and voltage. **b) and c)** Voltage and spike prediction performance of the respective surrogate models, produced using joint ablation of $d_m$ with $d_{\mathrm{mlp}} = 2d_m$ for ELM models. Previously around 10M parameters were required to make accurate spike predictions using a TCN Beniaguev et al. (2021), an LSTM baseline is able to do it with 266K, and our ELM and Branch-ELM neuron model require 53K and 8K respectively (3rd from left each), simultaneously achieving much better voltage prediction performance than the TCN. For comparison in terms of TP/FP Rate performance or FLOPS cost see Fig. 4c or S1 respectively. Additional comparisons to other phenomenological neuron models, such as LIF and ALIF, are provided in Table S4.

**Deep learning architectures for long sequence modeling** have seen a shift towards the explicit incorporation of timescales for improved temporal processing, as observed in recent advancements in RNNs, transformers, and state-space models Gu et al. (2021); Mahto et al. (2021); Smith et al. (2023); Ma et al. (2023). Such an explicit approach can be traced back to Leaky-RNNs Mozer (1991); Jaeger (2002); Kusupati et al. (2018); Tallec & Ollivier (2018), which use a convex combination of old memory and updates, as done in ELM using $\boldsymbol{\kappa_m}$. Whereas the classic time-varying memory decay mediated by implicit timescales Tallec & Ollivier (2018), is known from classic gated RNNs like LSTM Hochreiter & Schmidhuber (1997) and GRU Cho et al. (2014). In contrast to complex gating mechanisms, time-varying implicit timescales, or sophisticated large multi-staged architectures, the ELM features a much simpler recurrent cell architecture only using constant explicit (trainable) timescales $\boldsymbol{\tau_m}$ for gating, putting the major emphasis on the input integration dynamics using a single powerful MLP.

## 4  EXPERIMENTS

In the experimental section of this work, we address three primary research questions. **First,** can the ELM neuron accurately fit a high-fidelity biophysical simulation with a small number of parameters? We detail this investigation in Section 4.1. **Second,** how can the ELM neuron effectively integrate non-trivial temporal information? We explore this issue in Section 4.2. **Third,** what are the computational limits of the ELM design? Discussed in Section 4.3. For training details, hyper-parameters, and tuning recommendations, please refer to the Appendix Section B and Table S1.

### 4.1  FITTING A COMPLEX BIOPHYSICAL CORTICAL NEURON MODEL'S I/O RELATIONSHIP

The NeuronIO dataset primarily consists of simulated input-output (I/O) data for a complex biophysical layer 5 cortical pyramidal neuron model Hay et al. (2011). Input data features biologically inspired spiking patterns (1278 pre-synaptic spike channels featuring -1,1 or 0 as input), while output data comprises the model's somatic membrane voltage and output spikes (see Figure 2a and 2b). The dataset and related code are publicly available Beniaguev et al. (2021), and the models were trained using Binary Cross Entropy (BCE) for spike prediction and Mean Squared Error (MSE) for somatic voltage prediction, with equal weighting.

Our ELM neuron achieves better prediction of voltage and spikes than previously used architectures for any given number of trainable parameters (and compute). In particular, it crosses the "sufficiently good" spike prediction performance threshold (0.991 AUC) as proposed in (Beniaguev et al., 2021) by using 50K trainable parameters, which is around $200\times$ improvement compared to the previous attempt (TCN) that required around 10M trainable parameters, and $6\times$ improvement over a LSTM baseline which requires around 266K parameters (see Figure 2c-d). Overall, this result indicates that recurrent computation is an appropriate inductive bias for modeling cortical neurons.

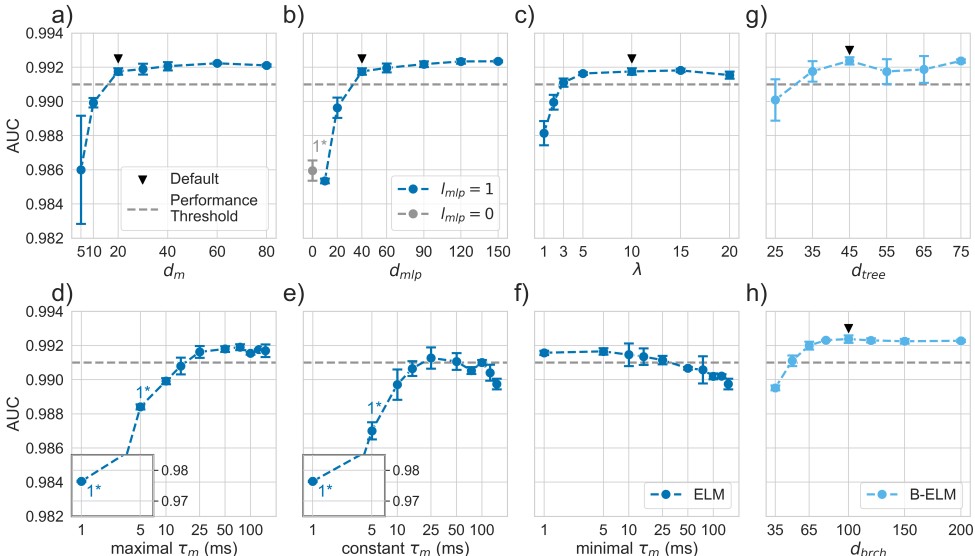

Figure 3: **The ELM neuron gives relevant neuroscientific insights.** Ablations on NeuronIO of different hyperparameters of an ELM neuron with AUC $\approx 0.992$, and a Branch-ELM with the same default hyperparameters. The number of removed divergent runs marked with $1^*$. **a)** We find between 10 and 20 memory-like hidden states to be required for accurate predictions, much more than typical phenomenological models use Izhikevich (2004); Dayan & Abbott (2005). **b)** Highly nonlinear integration of synaptic input is required, in line with recent neuroscientific findings Stuart & Spruston (2015); Jones & Kording (2022); Larkum (2022). **c)** Allowing greater updates to the memory units is beneficial (see Appendix A). **d-f)** Ablations of memory timescale (initialization and bounding) range or (constant) value, with the default range being 1ms-150ms. Timescales around 25 ms seems to be the most useful (matching the typical membrane timescale in the cortex Dayan & Abbott (2005)); however, a lack can be partially compensated by longer timescales. **g) and h)** Ablating the number of branches $d_{\text{tree}}$ and number of synapses per branch $d_{\text{brch}}$ of the Branch-ELM neuron.

We use the fitted model to investigate how many memory units and which timescales are needed to match the neuron closely. We find that around 20 memory units are required (Figure 3a) with timescales that are allowed to reach at least 25 ms (Figure 3d). While a diversity of timescales, including long ones, seems to be favorable for accurate modeling (Figure 3d and 3f), ELM with constant memory timescales around 25 ms performs sufficiently well (matching the typical membrane timescales in computational modeling Dayan & Abbott (2005), Figure 3e). Removing the hidden layer or decreasing the integration mechanism complexity significantly reduces performance (Figure 3b). Allowing for more rapid memory updates through larger $\lambda$ is crucial (Figure 3c), possibly to match the fast internal dynamics of neurons around spike times or to absorb information faster into memory (more details in Appendix A). When fitting the simple leaky-integrate-and-fire (LIF) or adaptive LIF, we reach a better prediction with only a few memory units (Figure S7).

**How much nonlinearity is in the dendritic tree?** Within the ELM architecture, we allow for nonlinear interaction between any two synaptic inputs via the MLP. This flexibility might be necessary in cases where little is *a priori* known about the input structure. However, for matching the I/O of cortical neurons, knowledge of neuronal morphology and biophysical assumptions about *linear-nonlinear* computations in the dendritic tree might be exploited to reduce the dimensionality of the input to the MLP (parameter-costly component with $d_s = 1278$ inputs). Consequently, we modify the ELM neuron to include virtual branches along which the synaptic input is first reduced by a simple summation before further processing (see Figure 4). For NeuronIO specifically, we assign the synaptic inputs to the branches in a moving window fashion (exploiting that in the dataset, neighboring inputs were also typically neighboring synaptic contacts on the same dendritic branch of the biophysical model). The window size is controlled by the branch size $d_{\text{brch}}$, and the stride size is derived from the number of branches $d_{\text{tree}}$ to ensure equally spaced sampling across the $d_s = 1278$ inputs.

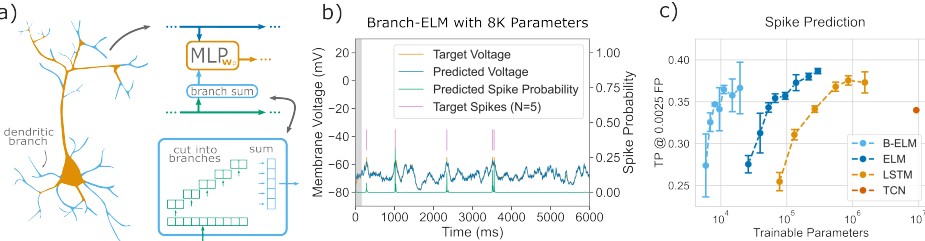

Figure 4: **Coarse-grained modeling of synaptic integration significantly improves model efficiency. a)** The integration mechanism dynamics of the ELM now computes the activity of individual dendritic branches as a simple sum of their respective synaptic inputs first before passing them on to the MLP$_{w_p}$, where $d_{\text{tree}}$ is the number of branches and $d_{\text{brch}}$ the number of synapses per branch. **b)** Accurate predictions using a Branch-ELM neuron with 8104 parameters (for zoomed-in version with model dynamics see Figure S9). **c)**. The new Branch-ELM neuron improves on the ELM neuron by about 7× in terms of parameter efficacy (same ELM hyper-parameters). Differences in model quality are highlighted when examining a True-Positive rate at a low False-Positive rate.

Surprisingly, even with this strong simplification, the Branch-ELM neuron model can retain its predictive performance while requiring 8K trainable parameters (roughly 7× reduction over the vanilla ELM) to cross the performance threshold substantially. We also find that a combination of $d_{\text{tree}} = 45$, $d_{\text{brch}} = 65$ and $d_m = 15$ still achieved over 0.9915 AUC with only 5329 trainable parameters, corroborating the assumption of the near-linear computation within dendritic branches and inviting future investigation of minimal required synaptic nonlinearity. However, this simplification utilizes the knowledge of morphology for modeling the neuron (in our case, exploiting the neighborhood in the dataset), violating it leads to deterioration of performance (Figure S6), therefore for most of the task we use the vanilla ELM neuron.

## 4.2 EVALUATING TEMPORAL PROCESSING CAPABILITIES ON A BIO-INSPIRED TASK

The Spiking Heidelberg Digits (SHD) dataset comprises spike-encoded spoken digits (0-9) in German and English Cramer et al. (2020). The digits were encoded using 700 input channels in a biologically inspired artificial cochlea. Each channel represents a narrow frequency band with the firing rate coding for the signal power in this band, resulting in an encoding that resembles the spectrogram of the spoken digit (see Figure 5a).

Motivated by recent findings that most neuromorphic benchmark datasets only require minimal temporal processing abilities Yang et al. (2021), we introduce the SHD-Adding dataset by concatenating two uniformly and independently sampled SHD digits and setting the target to their sum (regardless of language) (see Figure 5a). Solving this dataset necessitates identifying each digit on a shorter timescale and computing their sum by integrating this information over a longer timescale, which in turn requires retaining the first digit in memory. Whether single cortical neurons can solve this exact task is unclear; however, it has been shown that even single neurons possibly encode and perform basic arithmetics in the medial temporal lobe Cantlon & Brannon (2007); Kutter et al. (2018; 2022).

The ELM neuron solves the summing task across various temporal resolutions (determined by the bin size). As we vary the bin size from 1ms (2000 bins in total, the maximal temporal detail and longest required memory retention) to 100 (20 bins in total, the minimal temporal detail and shortest memory retention), the ELM neuron's performance remains robust, degrading proportionally to the bin sizes (see Figure 5b-d); this drop in performance is not a shortcoming of the model itself, but a consequence of loss of temporal information through binning. Further, the performance is also maintained when testing on two held-out speakers, showing that the ELM neuron remains comparatively robust out-of-distribution. Due to vanishing gradients, the LSTM performs worse on this task, especially when the bin size is below 50. As the bin size increases, the LSTM's performance improves but does not surpass ELM because larger bin sizes likewise lead to the loss of crucial temporal details. This outcome underlines the importance of a model's ability to integrate complex synaptic information effectively (see Figure 5e) and the utility of longer neuron-internal timescales for learning long-range dependencies, potentially necessary for cortical neuron's operation (see Figure 5f).

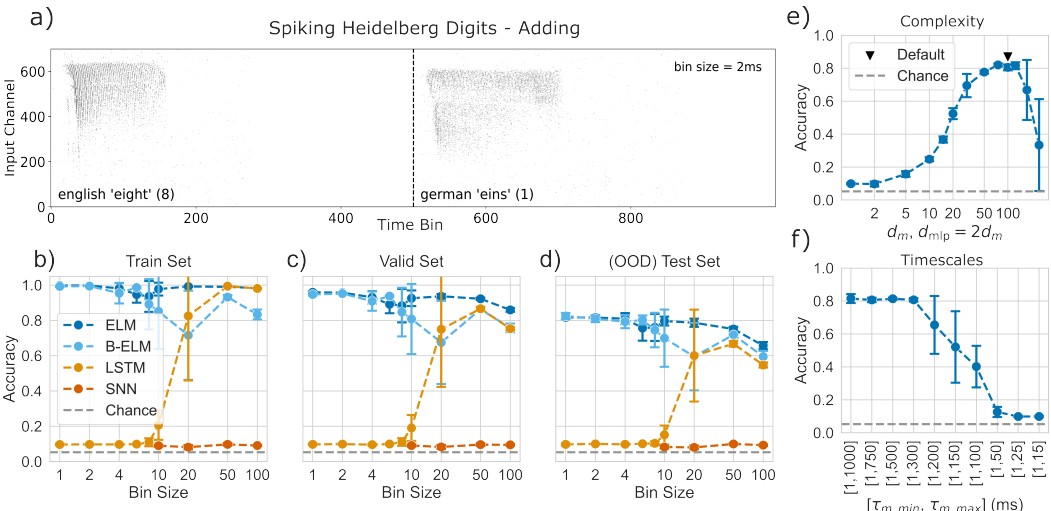

Figure 5: **The ELM neuron performs well on long and sparse data using longer timescales. a)** Sample from the biologically motivated SHD-Adding dataset (based on Cramer et al. (2020)), each dot is an input spike, and a vertical dashed line is a guide for the eye indicating the separation of the two digits (not communicated to the network). **b-d)** The ELM neuron (186K params.) consistently outperforms a classic LSTM (956K params.), especially for smaller bin sizes (meaning longer training samples), and LSTM-performance cannot be fully recovered even for larger bin sizes. The Branch-ELM (67K params.) can retain performance for fine-grained binning at a much reduced model size. Our LIF neuron based Spiking Neural Network (SNN) (51K params.) does not manage to achieve good performance for any bin size, and training becomes unstable for long sequences. **e) and f)** Ablations using a bin size of 2ms with test set performance reported. **e)** Solving SHD-Adding requires ELM neuron to have a higher complexity than required for NeuronIO, and much larger models become unstable. Potentially a network of smaller ELM neuron might be preferable. **f)** Longer $\tau_m$ are crucial for extracting long-range dependencies. Possibly shorter ones might suffice in a ELM network, as longer timescales can emerge through dynamics Khajehabdollahi et al. (2023).

## 4.3 EVALUATING ON COMPLEX AND VERY LONG TEMPORAL DEPENDENCY TASKS

To test the extent and limits of the ELM neuron's ability to extract complex long-range dependencies, we use the classic Long Range Arena (LRA) benchmark datasets Tay et al. (2021). It consists of classification tasks; three image-derived datasets Image, Pathfinder, and Pathfinder-X (images being converted to a grayscale pixel sequence), and three text-based datasets ListOps, Text, and Retrieval. Pixel and token sequences were encoded categorically, however, only considering 8 or 16 different grayscale levels for images. In particular, the Pathfinder-X task is notoriously difficult, as the task is to determine whether two dots are connected by a path in a $128 \times 128$ image (~16k length).

Our results are summarized in Table 1, where we compare the ELM neuron against several strong baselines. The model most comparable to ours is an LSTM with derived explicit gating bias initialization for effectively longer internal timescales Tallec & Ollivier (2018) (Chrono-LSTM). When comparing the two, we find that both models consistently perform well, except on the Pathfinder-X* task which only the ELM can reliably solve, albeit using longer $\tau_s$ than usual. The larger self-attention-based models trail further behind, with both Transformer Vaswani et al. (2017) and Longformer Beltagy et al. (2020) completely failing to solve the Pathfinder-X task Tay et al. (2021). Only the purpose-built architectures such as S4 Gu et al. (2021) and Mega Ma et al. (2023) (current SOTA) perform better, but they require many layers of processing and many more parameters than an ELM neuron, which uses 150 memory units and typically ~100k parameters.

Overall, the results suggest that the simple ELM neuron architecture is capable of reliably solving challenging tasks with very long temporal dependencies. Crucially, this required using memory timescales initialized according to the task length and highly nonlinear synaptic integration into 150

---

*Only once during hyper-parameter tuning did a single Chrono-LSTM run achieve barely above chance

Table 1: **The ELM neuron can solve challenging long-range sequence modeling tasks.** The table shows the mean accuracy on Long Range Arena (LRA) Benchmark Tay et al. (2021). The ELM neuron routinely scores higher than the Chrono-LSTM or the much larger Transformer or Longformer, and only the large multi-layered architectures tuned specifically for these tasks, such as S4 or Mega, outperform it. Surprisingly, it is also the only non purpose-built model that can reliably solve the notoriously challenging 16K sample length Pathfinder-X task. Model sizes of the bottom baseline models are extracted from Gu et al. (2021)Ma et al. (2023)Tay et al. (2021). Training details and model hyper-parameters are detailed in Appendix Section B, and Tables S2 and S3.

|  | Image | Pathfinder | Pathfinder-X | ListOps | Text | Retrieval |
|---|---|---|---|---|---|---|
| ELM Neuron (ours) | 49.62 | 71.15 | 77.29 | 46.77 | 80.3 | 84.93 |
| Chr.-LSTM Tallec & Ollivier (2018) | 46.09 | 70.79 | FAIL* | 44.55 | 75.4 | 82.87 |
| # parameters | ~100k | ~100k | ~100k | ~100k | ~200k | ~150k |
| Transformer Vaswani et al. (2017) | 42.44 | 71.4 | FAIL | 36.37 | 64.27 | 57.46 |
| Longformer Beltagy et al. (2020) | 42.22 | 69.71 | FAIL | 35.63 | 62.85 | 56.89 |
| S4 Gu et al. (2021) | 87.26 | 86.05 | 88.1 | 58.35 | 76.02 | 87.09 |
| Mega Ma et al. (2023) | 90.44 | 96.01 | 97.98 | 63.14 | 90.43 | 91.25 |
| # parameters | ~600k | ~600k | ~600k | ~600k | ~600k | ~600k |

memory units (See Appendix B). While the LRA benchmark revealed the single ELM neurons limits, we hypothesize that assembling ELM neurons into layered networks might give it enough processing capabilities to catch up with the deep models, but we leave this investigation to future work.

## 5 DISCUSSION

In this study, we introduced a biologically inspired recurrent cell, the Expressive Leaky Memory (ELM) neuron, and demonstrated its capability to fit the full spike-level input/output mapping of a high-fidelity biophysical neuron model (NeuronIO). Unlike previous works that achieved this fit with millions of parameters, a variant of our model only requires a few thousand, thanks to the careful design of the architecture exploiting appropriate inductive biases. Furthermore, unlike existing neuron models, the ELM can effectively model neuron without making rigid assumptions about the number of memory states and their timescales, or the degree of nonlinearity in its synaptic integration.

We further scrutinized the implications and limitations of this design on various long-range dependency datasets, such as a biologically-motivated neuromorphic dataset (SHD-Adding), and some notoriously challenging ones from the machine learning literature (LRA). Leveraging slowly decaying memory units and highly nonlinear dendritic integration into multiple memory units, the ELM neuron was found to be quite competitive, in particular, compared to classic RNN architectures like the LSTM, a notable feat considering its much simpler architecture and biological inspiration.

It should be noted that despite its biological motivation, our model cannot give mechanistic explanations of neural computations as biophysical models do, and that the task of fitting another neuron's I/O is not itself a biologically relevant task for a neuron. Many biological implementation details are abstracted away in favor of computational efficiency and conceptual insight, and the required/recovered ELM neuron hyper-parameters depend on what constitutes a sufficiently good fit and the model's subsequent use case. Furthermore, ELM is trained using BPTT, which is not considered biologically plausible in itself, and ELM learning likely relies on neuronal plasticity beyond synapses, the extent of which in biological neurons is still a subject of debate. Additionally, our neuron model dendrites are rudimentary (e.g., lacking apical vs basal distinction) and rely on oversampling synaptic inputs for performance so far. Finally, given the use of biologically implausible BPTT as a training technique and comparatively larger ELM neuron sizes on the later datasets, one should be careful to directly draw conclusions about the learning capabilities of individual biological cortical neurons.

Despite these caveats, the ELM's ability to efficiently fit cortical neuron I/O and its promising performance on machine learning tasks suggests that we are beginning to incorporate the inductive biases that drive the development of more intelligent systems. Future research focused on connecting smaller ELM neurons into larger networks could provide even more insights into the necessary and dispensable elements for building smarter machines.

ACKNOWLEDGMENTS

This work was supported by a Sofja Kovalevskaja Award from the Alexander von Humboldt Foundation. We acknowledge the support from the BMBF through the Tübingen AI Center (FKZ: 01IS18039A and 01IS18039B). AL, GM, and BS are members of the Machine Learning Cluster of Excellence, EXC number 2064/1 – Project number 39072764. AS would like to thank the Max Planck Society for their generous financial support throughout the project. We would like to thank Antonio Orvieto for help with table S6. We thank the Max Planck Computing and Data Facility (MPCDF) staff, as the majority of computations were performed on the HPC system Raven.

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

# Appendix

## A  IMPLEMENTATION DETAILS

All computations were performed using Python 3.9, and the following libraries were instrumental in our implementation: jax 0.3.14 (coupled with jaxlib 0.3.10 as a GPU back-end) for auto-grad and auto-vectorization; equinox 0.8.0, a jax-based neural network library; optax 0.1.3, a jax-based optimizer library; and pytorch 1.12.1 for data-loading. The accompanying git repository for the project can be found under: https://github.com/AaronSpieler/elmneuron.

Table S1: The ELM neuron parameters and recommendations.

| Hyper-Parameter | NeuronIO | Recommendation |
|---|---|---|
| $s_0$ | 0 | / |
| $w_s$ | 0.5 | / |
| $w_s$ (bounds) | $> 0$ | $>= 0$ |
| $\tau_s$ | all 5ms | $\propto$ time sparse data (tune) |
| $m_0$ | 0 | / |
| $d_m$ | 20 | up to 250 (primary TUNE) |
| $\tau_m$ (init technique) | equally spaced | evenly spaced on log scale |
| $\tau_m$ (init range) | 1ms, 100ms | 1ms, data length |
| $\tau_m$ (bounds) | 0ms, 500ms | same as init range |
| $\text{MLP}_{w_p}$ (nonlineary) | ReLU | / |
| $\text{MLP}_{w_p}$ (bias) | True | / |
| $\text{MLP}_{w_p}$ (init technique) | Kaiming Uniform | / |
| $l_{\text{mlp}}$ | 1 | / |
| $d_{\text{mlp}}$ | $2 * d_m$ | / |
| $\lambda$ | 10 | 5 |
| $w_y$ (bias) | True | / |
| $w_y$ (init technique) | Kaiming Uniform | / |
| $d_{\text{tree}}$ | 45 | $\propto$ inp. 2-5x sampled (tune) |
| $d_{\text{brch}}$ | 100 | $\propto$ space sparse data (tune) |

For all experiments $w_y$, $w_p$ and $\tau_m$ were learnable, with $w_s$ crucially also learnable for Branch-ELM.

**Recommended default and tuning parameters:**  We primarily recommend ablating $d_m$. In case of small $d_m$, exploring larger relative $d_{\text{mlp}}$ might yield improved performance. For ELM with many small $\tau_m$ or larger $\lambda$ we have observed spome trainig instability; seemingly resolved through modified memory update (see below). The timescales $\tau_m$ should generally be derived from the dataset length and the suspected timescales of the temporal dependencies within the data; if reasonably initialized, learnability doesn't seem to be necessary. Increasing $\tau_s$ may help to enhance learning speed in case of temporally very sparse data. When using the Branch-ELM it is important to sufficiently over-sample the input (more synapses than inputs) as we suspect significant expressivity stemming from $w_s$ doing the selection. Additional recommendations are summarized in Table S1.

**Timescale parametrization:**  The memory timescales $\tau_m$ are directly learnable model parameters. They are constrained to an apriori-specified bound, which is enforced through a $sigmoid$ rectification (the lower bound being $> 0$). By defining $\kappa_m = \exp(-\Delta t/\tau_m)$, the resulting values are ensured to be within $[0, 1]$ for all valid $\tau_m$, irrespective of $\Delta t$. In preliminary experiments we observed increased training stability as opposed to directly learning $\kappa_m$.

**Improved implementation:**  We found that enabling greater changes in $\Delta m_t$ by introducing a multiplicative factor $\lambda$ improved training performance. Interestingly, the $\lambda \cdot (1 - \kappa_m)$ term can then be seen as effectively using a $\lambda$ times faster input timescale than $\tau_m$. This notion can be explicitly implemented by substituting $\kappa_\lambda = \exp(-\Delta t/\tau_\lambda)$ where $\tau_\lambda = \tau_m/\lambda$. The appoximation holds for $\tau_m >> \lambda > 1$, and only diverges for small $\tau_m$, where it allows for less stark $m$ changes than the

original implementation. In preliminary experiments we found this modification to result in increased training stability for ELM neuron with many small $\tau_m$ or larger $\lambda$ (see Figure S4).

**Stable memory dynamics:**  A necessary prerequisite for solving long-range credit assignment problems in recurrent architectures is to address the vanishing and exploding gradient problem. In the ELM neuron we achieve this by enforcing a stable dynamic of the memory units, which are primarily responsible for carrying information forward through time. The combination of controlled (slow) decay using $\kappa_m$ (addressing the vanishing gradient problem) and controlled (bounded) growth using the complementary $1 - \kappa_m$ (addressing the exploding gradients problem), couples input to forget timescales using $\lambda$ in a principled way such that $m$ will be bounded if $\Delta m$ is bounded (e.g. ensured through $tanh$ rectification), even if latter was generated using a highly nonlinear MLP (see Figure S9d). Note that in principle, this construction could also be applied to layer wise processing in depth, instead of in recurrent processing in time; however, we leave this experiment to future investigations.

**The SNN implementation:**  The LIF neuron based SNN consisted of an output layer ($N_o$ = "number of classes") and a recurrent layer ($N_r = 500 - N_o$), with $20\%$ of neurons being inhibitory. The spiking threshold was $v_{thr} = 1$, and neurons were partially reset after firing using $v_{t+\Delta t} = v_t - 0.9 \cdot v_{thr}$. The membrane timescale was initialized to 25ms, and directly learnable like in the ELM neuron. Each of the 100 synaptic weights were initialized to $w_s = 0.3/sqrt(100)$, and were rectified using $ReLU$. All neurons were randomly connected on a synapse-by-synapse basis with $90\%$ probability to the previous-layer, and $10\%$ probability to the own-layer. The output neurons output was low-pass filtered using constant 20ms, before being used by the cross-entropy function.

# B  DATASETS AND TRAINING DETAILS

**General training setup:**  For each task and dataset, the training dataset was deterministically split to create a consistent validation dataset, which was used for model selection during training and hyperparameter tuning. All models were trained using Backpropagation Through Time (BPTT), and used a cosine-decay learning-rate schedule across the entire training duration of the training run. All experiments were run on a single A100-40GB or A100-80GB and ran less than 24h, Pathfinder-X being the notable exception.

**NeuronIO Dataset:**  For training and evaluation the dataset was pre-processed in accordance with Beniaguev et al. (2021), by capping somatic membrane voltage at -55mV and subtracting a bias of -67.7mV. Additionally, the somatic membrane voltage was scaled by 1/10 for training. Training samples were 500ms long with a 1ms bin size and $\Delta t$. The ELM neuron used the default parameters from Table S1. Models were trained using the Adam optimizer with an initial learning rate of $5e^{-4}$ and a batch size of 8 for 30 epochs with 11,400 batches per epoch using Binary Cross Entropy (BCE) for spike prediction and Mean Squared Error (MSE) for somatic voltage prediction, with equal weighting. Loss was calculated after a 150ms burn-in period. The mean and standard deviation over three runs is reported, with Root Means Squared Error (RMSE) and Area Under the Receiver Operator Curve (AUC) for voltage and spike prediction, respectively. The model hyper-parameters and training settings were chosen based on validation RMSE in preliminary ablations.

**Spiking Heidelberg Digits (Adding) Datasets:**  The digits were preprocessed by cutting them to a uniform length of one second and binning the spikes using various bin sizes, the default being 2ms. The models were trained using the Adamax optimizer with an initial learning rate of $5e^{-3}$ and a batch size of 8 for 70 epochs, with 814 or 2000 batches per epoch for SHD and SHD-Adding respectively, with $\Delta t$ set to the bin size, and dropout probability set to 0.5. The ELM and Branch-ELM used $\lambda = 5$, $d_m = 100$ and $\tau_m$ initialized evenly spaced between 1ms and 150ms with bounds of 0ms to 1000ms, whereas the LSTM used a hidden size of 250 and additional recurrent dropout of 0.3, while the SNN used a learning rate of $2e^{-3}$ no dropout but a $l1$ regularization on the spikes of 0.01. The Branch-ELM over-sampled the input with $d_{\text{tree}} = 100$, $d_{\text{brch}} = 15$ and used random synapse to branch assignment. Models were trained using the Cross-Entropy (CE) loss on the last float output of the respective model, and the performance was reported as prediction Accuracy, with mean and standard deviation calculated over five runs (chance performance being 1/19). The model hyper-parameters and training settings were chosen based on validation Accuracy in preliminary ablations.

**Long Range Arena Benchmark:** The images-based datasets were preprocessed by binning the individual grey-scale values (256 total) into 16, or 8 for Pathfinder-X, different levels. For the text based datasets a simple one-hot token encoding was used. For the Retrieval task with a two-tower setup, the latent-dimension was 75 for both models. All models were trained using the Adam optimizer, with the ELM neuron using an initial learning rate of $2e-4$, and the the LSTM models working best with $1e-3$. All were trained using Cross-Entropy (CE) loss on the last output of the model, and the performance is reported as prediction Accuracy. The mean over three runs is reported for all experiments.

Table S2: The ELM neuron configuration

| Dataset | Input Dim | Batch Size | Epochs | $\tau_s$ | Timescales |
|---------|-----------|------------|--------|----------|------------|
| Image | 16 | 384 | 300 | 5 | logspace: $1-10^3$ |
| Pathfinder | 16 | 384 | 300 | 5 | logspace: $1-10^3$ |
| Pathfinder-X | 8 | 768 | 300 | 150 | logspace: $1-2*10^4$ |
| ListOps | 25 | 384 | 150 | 5 | logspace: $1-2*10^3$ |
| Text | 169 | 384 | 150 | 5 | logspace: $1-4*10^3$ |
| Retrieval | 105 | 384 | 150 | 5 | logspace: $1-4*10^3$ |

The ELM memory timescale bounds were matched to the initialization range. The ELM used a synapse tau of $150ms$ on the Pathfinder-X dataset, which we observed to increase the learning speed significantly, however, smaller synapse tau can also work (e.g. see Figure S8). Otherwise, hyper-parameters were harmonized as much as possible, to demonstrate the robustness of the hyper-parameter choice.

Table S3: The Chrono-LSTM configuration

| Dataset | Input Dim | Batch Size | Epochs | Timescales |
|---------|-----------|------------|--------|------------|
| Image | 16 | 384 | 300 | uniform: $1-2*10^3$ |
| Pathfinder | 16 | 384 | 300 | uniform: $1-2*10^3$ |
| Pathfinder-X | 8 | 768 | 300 | uniform: $1-2*10^4$ |
| ListOps | 25 | 384 | 150 | uniform: $1-4*10^3$ |
| Text | 169 | 384 | 150 | uniform: $1-8*10^3$ |
| Retrieval | 105 | 384 | 150 | uniform: $1-8*10^3$ |

The Chrono-LSTM hyper-parameter tuning primarily concerned the learning rates and hidden sizes, however a learning rate of $1e^{-3}$ (among the tested) and a hidden size of 150 (max tested) consistently performed best, the exception being Pathfinder-X, where during tuning a single run using a smaller hidden size performed slightly above change.

## C  ADDITIONAL RESULTS

For comparison, we fit the classic computational neuroscience models to the same NeuronIO data. Namely, we use the Leaky integrate-and-fire (LIF) neuron model (with learnable membrane timescale, weights, and bias unit for linear integration), adaptive LIF (ALIF) that has additionally a single timescale of spike-frequency adaptation. To have a fair comparison, we also fitted an ELM neuron model with only a single memory unit (and timescale) with linear synaptic integration (thus no MLP, additional parameters due to bias units, readout, and other implementation details). The LIF's internal membrane voltage was directly fit to the target voltage, and its output spike directly to the target spikes, using otherwise same training methodology as described in Section B.

Table S4: Evaluating Tiny Models on NeuronIO

| Model | Soma RMSE | Spike AUC | # parameters |
|---|---|---|---|
| (classic) LIF | 2.29 | 0.9117 | 1280 |
| (classic) ALIF | 2.29 | 0.9115 | 1281 |
| (simplest) ELM | 1.60 | 0.9255 | 1286 |
| ($d_m = 15$) Branch-ELM | 0.66 | 0.9915 | 5329 |

While all three models perform much worse than the reference performance threshold of $0.991AUC$, the ELM neuron performs slightly better, particularly for somatic prediction. This could result from the ELM neuron, similar to the underlying biophysical model, not enforcing an explicit hard memory reset when spiking. The noticeable lack of performance difference between LIF and ALIF might be due to the fitted models consistently staying below the spiking threshold. Finally, the LIF and ALIF model's shortcoming in accurately capturing the I/O relationship highlights the need for a more flexible phenomenological neuron model.

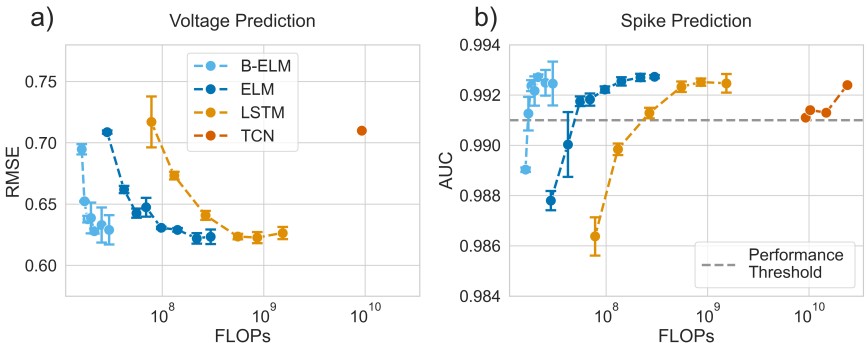

Figure S1: **The ELM neuron is a computationally efficient model of cortical neuron.** Similar figure to 2c and 2d, however, displaying FLOPs required to do inference on a single sample. **a) and b)** Voltage and spike prediction performance of the respective surrogate models. While previous works required around 10M parameters to make accurate spike predictions using a TCN Beniaguev et al. (2021), an LSTM baseline is able to do it with 266K, our ELM neuron model requires merely 53K, and our Branch-ELM neuron only a humble 8K, simultaneously achieving much better voltage prediction performance than the TCN. A throughput optimized ELM neuron implementation can potentially reduce the required FLOPs even further.

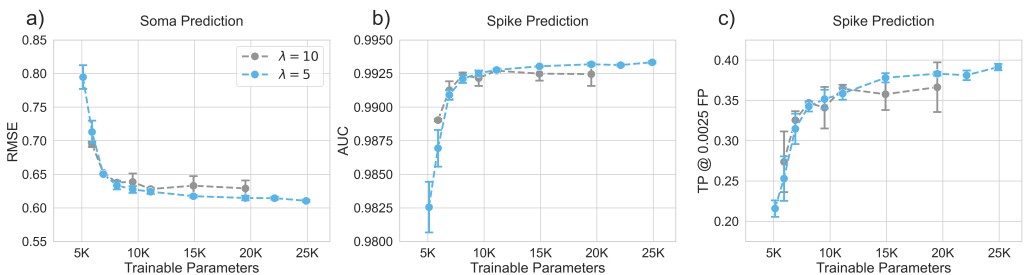

Figure S2: **Branch-ELM neuron training is more stable with smaller $\lambda$.** Evaluating a Branch-ELM with same hyper-parameters as in Figure 2, except with $\lambda = 5$. The variability of test set performance is reduced for most configurations, particularly for ones with a larger number of trainable parameters (and memory units). Additionally, it allows for an improved max performance for the model type.

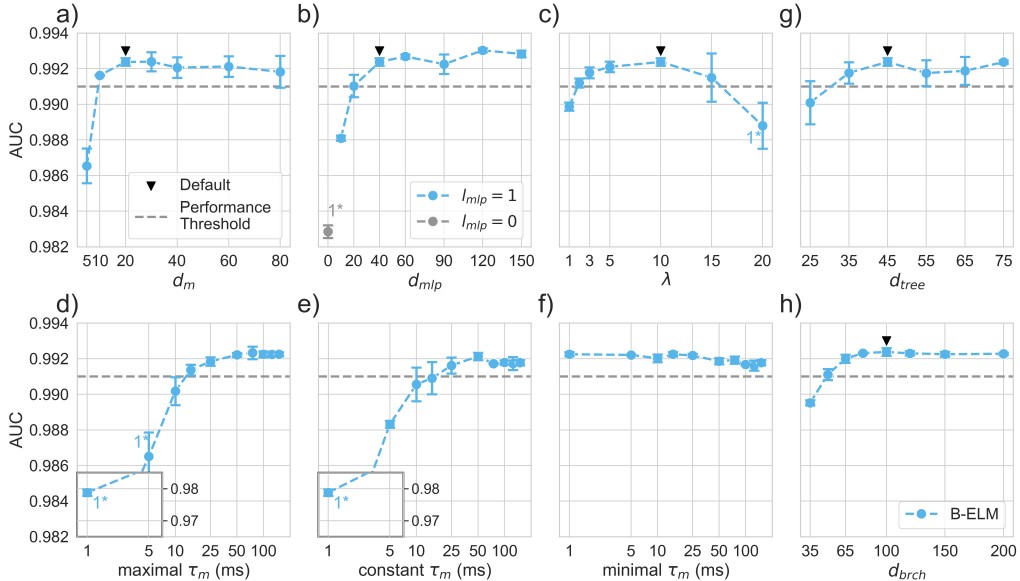

Figure S3: **The ELM neuron gives relevant neuroscientific insights.** Ablations on NeuronIO of different hyperparameters of an Branch-ELM neuron with AUC ≈ 0.992, with default hyperparameters. The number of removed divergent runs marked with $1^*$. **a)** We find above memory-like hidden states to be required for accurate predictions, much more than typical phenomenological models use Izhikevich (2004); Dayan & Abbott (2005). **b)** Highly nonlinear integration of synaptic input is required, in line with recent neuroscientific findings Stuart & Spruston (2015); Jones & Kording (2022); Larkum (2022). **c)** Allowing greater updates to the memory units is beneficial, however, too large ones increase training instability. **d-f)** Ablations of memory timescale (initialization and bounding) range or (constant) value, with the default range being 1ms-150ms. Timescales around 25ms-50ms seem to be the most useful (matching the typical membrane timescale in the cortex Dayan & Abbott (2005)); however, a lack can be partially compensated by longer timescales, even better than by the vanilla ELM. **g) and h)** Ablating the number of branches $d_{\text{tree}}$ and number of synapses per branch $d_{\text{brch}}$.

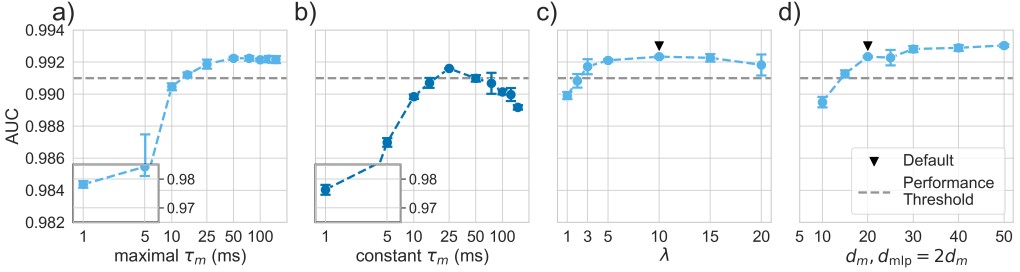

Figure S4: **Ablations with the improved ELM neuron implementation.** Ablations on NeuronIO that previously displayed training instabilities are now stable throughout and more consistent when rerun with the updated implementation (see section A for details on the implementation). **a)** Rerun of experiment in S3d. **b)** Rerun of experiment in 3e. **c)** Rerun of experiment in S3c. **d)** Rerun of experiment in 2c. Furthermore, we reran experiment in 3b, and training was stable for linear integration. Lastly, we reran experiment in 5e, however, did not observe significant improvements.

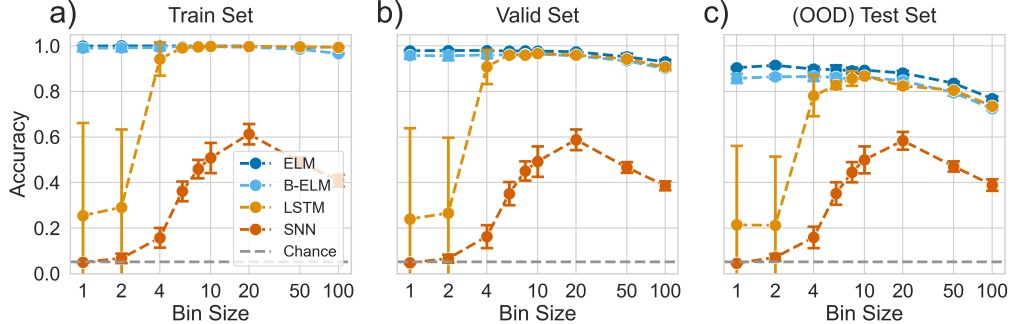

Figure S5: **The ELM neuron performs well on typical neuromorphic datasets.** The following results are on the original Spiking Heidelberg Digits dataset Cramer et al. (2020). **a-c)** The ELM and Branch-ELM neuron reliably outperforms a classic LSTM, especially for smaller bin sizes (meaning longer training samples), and LSTM-performance cannot be fully recovered even for larger bin sizes. Our LIF neuron based Spiking Neural Network (SNN), however, does manage to achieve decent performance for bin sizes around 20, in contrast to the SHD-Adding dataset (see Figure 5).

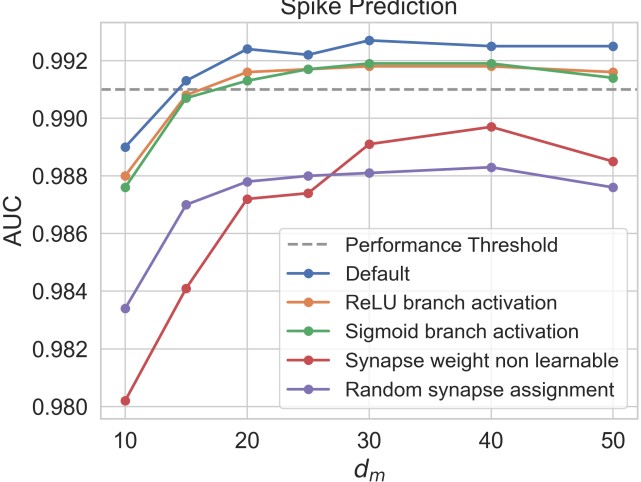

Figure S6: **Ablating the ELM branch architecture on NeuronIO.** Average test AUC displayed, using otherwise same hyper-parameters and training setup as in experiments in Figure 2. Exploiting the ordering in the synaptic input, and having learnable synapses is crucial for the Branch-ELM neuron model. Applying a specific nonlinearity on branch output slightly degrades performance.

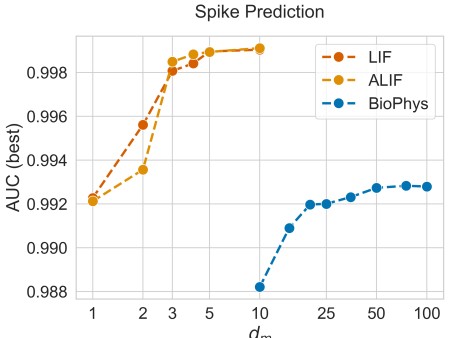

Figure S7: **Fitting simplified neuron models using ELM.** Accurately fitting the classic Leaky Integrate and Fire (LIF) model is possible with only linear integration ($l_{mlp} = 0$), and a single memory unit ($d_m = 1$), therefore matching the ground truth architecture. When fitting both LIF and Adaptive-LIF with an ELM neuron with two memory units, the LIF fit yields better results; this is expected, as the ground truth ALIF architecture has an additional hidden state; the adaptive threshold. We suspect that as neither LIF nor ALIF display chaotic dynamics, an overall higher AUC may be achieved than for BioPhys; the AUC plateau may display residual uncertainty inherent to the dataset.

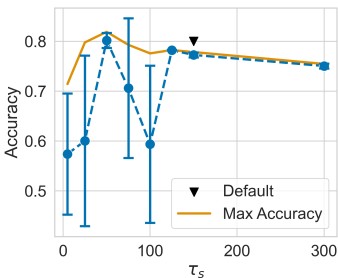

Figure S8: **Ablating the $\tau_s$ parameter on Pathfinder-X.** Average and max test Accuracy displayed, using otherwise same hyper-parameters and training setup as in experiments in Table S2. While reliably achieving high performance requires larger $\tau_s$, smaller timescales can achieve even higher performance, although less reliably and take longer to pick up the learning signal. The intermediate drop in performance for 75ms and 100ms could be an artifact due to nontrivial interactions between $\tau_s$ and the cycle length (128) of the flattened image ($128 \times 128$) data.

Table S5: Ablation of ELM neuron $d_m$ on Pathfinder

| $d_m$ | 10 | 25 | 50 | 100 | 150 | 200 | 300 |
|---|---|---|---|---|---|---|---|
| # params | 822 | 3552 | 12k | 44k | 96k | 168k | 373k |
| ACC | 57.83 | 62.95 | 66.31 | 69.31 | 71.54 | 72.98 | 71.85 |

In Table S5 we show the dependence of test accuracy on the ELM neuron model size, using otherwise same training setup as before. Performance levels out around $72\%$ accuracy at $d_m = 150$ (default), and decreases to $57\%$ accuracy at $d_m = 10$. An S5 model (see Table S6 with likewise a single layer (and $186K$ parameters) is outperformed with an ELM neuron with $d_m = 25$ memory units (and $3.5K$ parameters).

Table S6: Ablation of S5 model layers on Pathfinder

| # layers | 1 | 2 | 4 | 6 |
|---|---|---|---|---|
| # params | 186k | 371k | 742k | 1.1M |
| ACC | 58.19 | 78.69 | 91.63 | 95.33 |

In Table S6 provide an ablation of the S5 model Smith et al. (2023), a close to state of the art model on the Long Range Arena, as reference of how such models perform with varying number of parameters and layers. The reported training hyper-parameters were used, with the learning rate individually ablated per model size. The mean accuracy over three runs is reported. Note, the steep drop-off in performance between two layers and one.

Table S7: Ablation of ELM neuron dropout probability on Pathfinder

| dropout | 0 | 0.1 | 0.2 | 0.3 | 0.4 | 0.5 |
|---|---|---|---|---|---|---|
| Train ACC | 0.8525 | 0.9092 | 0.9244 | 0.9149 | 0.8987 | 0.8763 |
| Test ACC | 0.7275 | 0.7631 | 0.811 | 0.823 | 0.815 | 0.8063 |

In Table S7 provide an ablation of the ELM neuron training with varying dropout probability and longer training (1000 Epochs), but otherwise same hyper-parameters as before ($d_m = 150$). Notice, how using a dropout of 0.3 outperforms even two layered S5 at less than a third of its size. Given the significant gap between train and test accuracy, we expect that further improvements to the training setup by using weight decay, layer normalization, etc. (as routinely used in the training of SOTA models on LRA) might improve convergence speed and generalization.

## D    ADDITIONAL VISUALIZATIONS

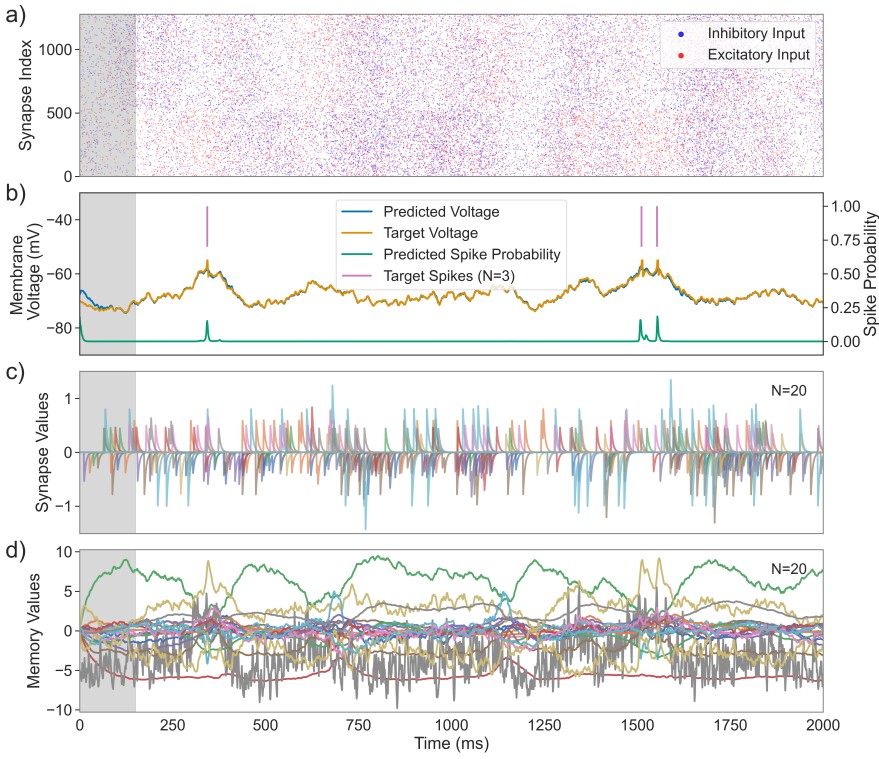

Figure S9: **Visualization of ELM neuron dynamics.** Extended visualization of Figure 2b for an ELM neuron achieving around 0.992 AUC. **a)** The synaptic input as the neuron model receives it, with excitatory input ($+1$) marked in red, and inhibitory input ($-1$) marked in blue. **b)** The ELM neurons predictions and the ground-truth targets for a regular sample from the data. Interestingly, the whole two seconds were inferred in one go (similar to Figure 4b), which shows its generalization capabilities beyond the training horizon of 500ms. **c)** A random subset of 20 synapses are visualized. Synapses receiving negative input will be deflected downwards. **d)** All 20 memory values are visualized. Some fluctuate more rapidly than others, typically proportional to their memory timescales.

# E  Results in Table Format

Table S8: The Branch-ELM on NeuronIO

| Memory Units | Trainable Parameters | FLOPs | Voltage Prediction RMSE | Spike Prediction AUC |
|---|---|---|---|---|
| 10 | 5.9K | 16.04M | 0.695 ± 0.004 | 0.989 ± 0.0001 |
| 15 | 6.9K | 17.06M | 0.652 ± 0.0 | 0.9913 ± 0.0007 |
| 20 | 8.1K | 18.27M | 0.638 ± 0.003 | 0.9924 ± 0.0002 |
| 25 | 9.5K | 19.69M | 0.639 ± 0.013 | 0.9922 ± 0.0006 |
| 30 | 11.1K | 21.3M | 0.628 ± 0.002 | 0.9927 ± 0.0001 |
| 40 | 14.9K | 25.13M | 0.633 ± 0.014 | 0.9925 ± 0.0005 |
| 50 | 19.5K | 29.76M | 0.629 ± 0.012 | 0.9925 ± 0.0009 |

Table S9: The ELM on NeuronIO

| Memory Units | Trainable Parameters | FLOPs | Voltage Prediction RMSE | Spike Prediction AUC |
|---|---|---|---|---|
| 10 | 26.06K | 28.49M | 0.709 ± 0.001 | 0.9878 ± 0.0004 |
| 15 | 39.39K | 41.84M | 0.662 ± 0.003 | 0.99 ± 0.0013 |
| 20 | 52.92K | 55.38M | 0.643 ± 0.004 | 0.9918 ± 0.0002 |
| 25 | 66.65K | 69.13M | 0.648 ± 0.008 | 0.9918 ± 0.0002 |
| 35 | 94.71K | 97.22M | 0.631 ± 0.001 | 0.9922 ± 0.0001 |
| 50 | 138.3K | 140.85M | 0.629 ± 0.002 | 0.9925 ± 0.0002 |
| 75 | 214.95K | 217.58M | 0.622 ± 0.004 | 0.9927 ± 0.0001 |
| 100 | 296.6K | 299.3M | 0.623 ± 0.006 | 0.9927 ± 0.0001 |

Table S10: The LSTM on NeuronIO

| Hidden Size | Trainable Parameters | FLOPs | Voltage Prediction RMSE | Spike Prediction AUC |
|---|---|---|---|---|
| 15 | 77.67K | 77.77M | 0.717 ± 0.021 | 0.9864 ± 0.0008 |
| 25 | 130.45K | 130.61M | 0.673 ± 0.003 | 0.9898 ± 0.0002 |
| 50 | 265.9K | 266.23M | 0.641 ± 0.004 | 0.9913 ± 0.0002 |
| 100 | 551.8K | 552.45M | 0.624 ± 0.002 | 0.9923 ± 0.0002 |
| 150 | 857.7K | 858.68M | 0.623 ± 0.005 | 0.9925 ± 0.0001 |
| 250 | 1529.5K | 1531.13M | 0.626 ± 0.005 | 0.9925 ± 0.0004 |

Table S11: SHD Results

| Model | Bin Size | Train Accuracy | Valid Accuracy | Test Accuracy |
|---|---|---|---|---|
| B-ELM | 1 | 0.99 ± 0.01 | 0.96 ± 0.01 | 0.86 ± 0.02 |
| | 2 | 0.99 ± 0.01 | 0.96 ± 0.02 | 0.86 ± 0.01 |
| | 4 | 0.99 ± 0.01 | 0.96 ± 0.01 | 0.86 ± 0.02 |
| | 6 | 1.0 ± 0.0 | 0.96 ± 0.0 | 0.86 ± 0.01 |
| | 8 | 1.0 ± 0.0 | 0.96 ± 0.0 | 0.85 ± 0.01 |
| | 10 | 1.0 ± 0.0 | 0.96 ± 0.0 | 0.86 ± 0.01 |
| | 20 | 0.99 ± 0.0 | 0.96 ± 0.01 | 0.85 ± 0.01 |
| | 50 | 0.99 ± 0.0 | 0.93 ± 0.0 | 0.79 ± 0.01 |
| | 100 | 0.97 ± 0.0 | 0.9 ± 0.01 | 0.72 ± 0.01 |
| ELM | 1 | 1.0 ± 0.0 | 0.98 ± 0.0 | 0.9 ± 0.0 |
| | 2 | 1.0 ± 0.0 | 0.98 ± 0.0 | 0.91 ± 0.01 |
| | 4 | 1.0 ± 0.0 | 0.98 ± 0.0 | 0.9 ± 0.01 |
| | 6 | 1.0 ± 0.0 | 0.98 ± 0.0 | 0.9 ± 0.02 |
| | 8 | 1.0 ± 0.0 | 0.98 ± 0.0 | 0.89 ± 0.01 |
| | 10 | 1.0 ± 0.0 | 0.98 ± 0.0 | 0.89 ± 0.0 |
| | 20 | 1.0 ± 0.0 | 0.97 ± 0.0 | 0.88 ± 0.01 |
| | 50 | 0.99 ± 0.01 | 0.95 ± 0.01 | 0.84 ± 0.01 |
| | 100 | 1.0 ± 0.0 | 0.93 ± 0.01 | 0.77 ± 0.01 |
| LSTM | 1 | 0.26 ± 0.41 | 0.24 ± 0.4 | 0.21 ± 0.35 |
| | 2 | 0.29 ± 0.34 | 0.27 ± 0.33 | 0.21 ± 0.3 |
| | 4 | 0.94 ± 0.07 | 0.91 ± 0.07 | 0.78 ± 0.09 |
| | 6 | 0.99 ± 0.0 | 0.96 ± 0.01 | 0.83 ± 0.02 |
| | 8 | 1.0 ± 0.0 | 0.96 ± 0.0 | 0.86 ± 0.03 |
| | 10 | 1.0 ± 0.0 | 0.97 ± 0.0 | 0.87 ± 0.01 |
| | 20 | 1.0 ± 0.0 | 0.96 ± 0.0 | 0.82 ± 0.01 |
| | 50 | 1.0 ± 0.0 | 0.94 ± 0.01 | 0.81 ± 0.0 |
| | 100 | 0.99 ± 0.0 | 0.91 ± 0.01 | 0.74 ± 0.0 |
| SNN | 1 | 0.05 ± 0.0 | 0.05 ± 0.0 | 0.05 ± 0.0 |
| | 2 | 0.07 ± 0.02 | 0.07 ± 0.02 | 0.07 ± 0.02 |
| | 4 | 0.16 ± 0.04 | 0.16 ± 0.05 | 0.16 ± 0.05 |
| | 6 | 0.36 ± 0.04 | 0.35 ± 0.05 | 0.35 ± 0.05 |
| | 8 | 0.46 ± 0.04 | 0.45 ± 0.04 | 0.45 ± 0.04 |
| | 10 | 0.51 ± 0.07 | 0.49 ± 0.07 | 0.5 ± 0.06 |
| | 20 | 0.61 ± 0.04 | 0.59 ± 0.05 | 0.58 ± 0.04 |
| | 50 | 0.49 ± 0.02 | 0.47 ± 0.02 | 0.47 ± 0.02 |
| | 100 | 0.41 ± 0.03 | 0.39 ± 0.02 | 0.39 ± 0.03 |

Table S12: SHD-Adding Results

| Model | Bin Size | Train Accuracy | Valid Accuracy | Test Accuracy |
|---|---|---|---|---|
| B-ELM | 1 | 0.99 ± 0.0 | 0.95 ± 0.01 | 0.83 ± 0.02 |
| | 2 | 0.99 ± 0.0 | 0.95 ± 0.0 | 0.81 ± 0.02 |
| | 4 | 0.95 ± 0.06 | 0.91 ± 0.05 | 0.79 ± 0.04 |
| | 6 | 0.99 ± 0.0 | 0.94 ± 0.0 | 0.8 ± 0.03 |
| | 8 | 0.89 ± 0.14 | 0.85 ± 0.14 | 0.75 ± 0.1 |
| | 10 | 0.85 ± 0.21 | 0.81 ± 0.2 | 0.7 ± 0.16 |
| | 20 | 0.72 ± 0.25 | 0.68 ± 0.24 | 0.59 ± 0.19 |
| | 50 | 0.93 ± 0.01 | 0.86 ± 0.01 | 0.72 ± 0.0 |
| | 100 | 0.83 ± 0.03 | 0.76 ± 0.02 | 0.59 ± 0.03 |
| ELM | 1 | 1.0 ± 0.0 | 0.96 ± 0.01 | 0.82 ± 0.01 |
| | 2 | 1.0 ± 0.0 | 0.96 ± 0.0 | 0.82 ± 0.01 |
| | 4 | 0.98 ± 0.03 | 0.93 ± 0.03 | 0.81 ± 0.03 |
| | 6 | 0.95 ± 0.05 | 0.89 ± 0.05 | 0.76 ± 0.07 |
| | 8 | 0.94 ± 0.09 | 0.88 ± 0.09 | 0.76 ± 0.08 |
| | 10 | 0.98 ± 0.04 | 0.93 ± 0.04 | 0.8 ± 0.03 |
| | 20 | 0.99 ± 0.0 | 0.94 ± 0.01 | 0.79 ± 0.02 |
| | 50 | 0.99 ± 0.0 | 0.92 ± 0.01 | 0.75 ± 0.01 |
| | 100 | 0.98 ± 0.0 | 0.86 ± 0.01 | 0.66 ± 0.02 |
| LSTM | 1 | 0.1 ± 0.0 | 0.1 ± 0.0 | 0.1 ± 0.01 |
| | 2 | 0.1 ± 0.0 | 0.1 ± 0.0 | 0.1 ± 0.0 |
| | 4 | 0.1 ± 0.0 | 0.1 ± 0.0 | 0.1 ± 0.0 |
| | 6 | 0.1 ± 0.0 | 0.1 ± 0.0 | 0.1 ± 0.0 |
| | 8 | 0.11 ± 0.02 | 0.11 ± 0.02 | 0.1 ± 0.01 |
| | 10 | 0.21 ± 0.08 | 0.19 ± 0.07 | 0.15 ± 0.05 |
| | 20 | 0.83 ± 0.37 | 0.75 ± 0.33 | 0.6 ± 0.26 |
| | 50 | 0.99 ± 0.0 | 0.87 ± 0.01 | 0.67 ± 0.02 |
| | 100 | 0.98 ± 0.0 | 0.75 ± 0.01 | 0.55 ± 0.01 |
| SNN | 10 | 0.09 ± 0.01 | 0.09 ± 0.01 | 0.08 ± 0.01 |
| | 20 | 0.08 ± 0.01 | 0.08 ± 0.01 | 0.08 ± 0.01 |
| | 50 | 0.1 ± 0.01 | 0.1 ± 0.01 | 0.1 ± 0.0 |
| | 100 | 0.09 ± 0.0 | 0.09 ± 0.0 | 0.09 ± 0.0 |

