# OpenReview forum: "The Expressive Leaky Memory Neuron: an Efficient and Expressive Phenomenological Neuron Model Can Solve Long-Horizon Tasks."
_ICLR.cc/2024/Conference — ICLR 2024 poster_

### Official Review · Reviewer_8X2o · 2023-10-26

**Soundness:** 2 fair
**Presentation:** 3 good
**Contribution:** 2 fair
**Rating:** 5
**Confidence:** 4

**Summary:**

In this work, the authors propose a model of neurons called Expressive Leaky Memory (ELM) (and Branch ELM later introduced in the paper). The goal is to build a model with fewer parameters than existing models that can still learn to replicate the input/output relationship of some pyramidal neurons. The authors also extend the evaluation to other tasks that are less neuroscience-inspired but aim at evaluating the ability of the model to capture long-term dependencies in the dataset.

The model is reminiscent of an LSTM with significant modifications leading to the possibility of better “synaptic” integration and longer timescales.

The authors do not aim to build a biologically plausible model of pyramidal neurons but to replicate some of its capabilities using less learnable parameters than existing literature, mostly other LSTMs and a TCN.

**Strengths:**

The paper's strengths are the following.

The paper is well-written and the task at hand is very interesting. As pointed out by the authors, not many models exist that can effectively represent the IO relationship of pyramidal neurons, and being able to do that could lead to improvement for various machine learning problems.

The presentation of numerical simulations and the exploration of machine learning tasks beyond neuroscience are commendable.

**Weaknesses:**

The weakness of the paper.

Although the paper argues that “merely” a few thousand learnable parameters is a good way to replicate what a single pyramidal neuron performs, it seems to still be a lot to me. In my understanding, the paper doesn’t address the task that the pyramidal neurons would perform, and thus, it is not clear such a model would be able to learn the same input/output relationship if only inputs were given.

Various papers, including recent papers on predictive coding and canonical correlation analysis, have suggested possible tasks performed by some pyramidal neurons. It would be interesting to see if such a model of neurons could learn the same IO based on this learning paradigm.

As pointed out, the synaptic plasticity and how the model's training is performed are not addressed.

Regarding the numerical experiments, the model is only compared to other “modern” ML models and not those that are more biologically plausible. I would like to see where simple pyramidal neuron models perform on spiking methods such as the one presented in this work. Compared to LSTMs and TCNs, it is not merely as relevant.

The term inductive bias is often used in papers when trying to characterize the “reasonable” choice of architecture as being inspired by biological facts. I can appreciate that the term is currently “hype,” but it is possibly misleading when considered at a machine learning conference where inductive bias means something else. The term hand-engineered would be more appropriate here.

Although the paper is well written, the choice of wording in many places is unscientific, e.g., “struggle to learn at all,” “merely a few thousand,” “merely meant to capture,” “degrading only gracefully,” and more. I would appreciate it if the authors paid more attention to possible bias in the writing of the paper.

The introduction of the Branch-ELM in Figure 4 appears too late in the paper. We are introduced to the concept at the same time as the results of the experiments when it would have been better to have it in Section 2 when the ELM is introduced.


In conclusion, I believe that the paper has some value, but I am not certain that it is well-suited for the venue. I also believe the paper doesn’t deliver on the claims made in the abstract or at the end of the introduction. I believe that what is achieved in the paper is overstated. Also, I would like to get the authors to write the paper with less biased words, as was mentioned in the weakness section.

**Questions:**

Based on the weaknesses highlighted above, I would suggest the authors address how they would provide results that more closely align with the claims. And provide some improvement on the various fronts that I have highlighted.

---

> ### Author Response · Authors · 2023-11-20
>
> We thank the referee for evaluating our manuscript and for constructive criticism, in particular concerning scientific writing style, earlier introduction of Branch-ELM, and overall limitations of the work.
>
> 1) *Using more scientific and neutral wording for clarity:*
>
> We agree the manuscript can be improved by replacing and reformulating ambiguous wording (such as the ones mentioned above) to be more scientific and neutral. We updated the manuscript where we found such formulations accordingly. Furthermore, we also introduce the Branch-ELM together with the ELM at the beginning of the manuscript.
>
> 2) *Aligning the abstract and contributions formulations for transparency:*
>
> Likewise, we concede that some sentences can be improved, we would like to offer the following modification, but are open to further changes to increase alignment:
>
> Abstract modifications:
>
> - Replace the phrase “the ELM neuron proves to be competitive on both datasets” with “the ELM neuron displays substantial long-range processing capabilities”
> - Replace the last sentence with “These findings raise further questions about the computational sophistication of individual cortical neurons and their role in extracting long-range temporal dependencies.”
>
> Contribution modifications:
>
> - 3rd point: The ELM neuron facilitates the formulation and validation of hypotheses regarding the underlying high-level neuronal computations using suitable architectural ablations.
> - 4th point: Lastly, we demonstrate the considerable long-sequence processing capabilities of the ELM neuron through the use of long memory and synapse timescales.
>
> 3) *Other neuron models on NeuronIO and their sizes:*
>
> The number of trainable parameters might look deceptively large, but the input dimension is also very large (1278 on the NeuronIO). To make a proper comparison, we followed your suggestion and fitted additional models to the NeuronIO dataset (Table S4, also referenced now in the text). Our simplest model, a Leaky Integrate and Fire (LIF) neuron, has 1280 parameters and achieves 0.9117 AUC, and an Adaptive-LIF (ALIF) neuron, has 1281 and achieves 0.9115 AUC. In comparison, we have found that a Branch-ELM with 5329 parameters ($dm=15$, $d_{tree}=45$ and $d_{brch}=65$) can achieve 0.9915 AUC. Thus, a Branch-ELM neuron with about 4 parameters per input dimension  achieves a good fit to the data.
>
> 4) *Fitting other neuron models to expand neuroscience contribution:*
>
> We also additionally provide results on fits of ELM neurons to LIF and ALIF data (Figure S6). The ELM neuron provides accurate fits with linear integration and few memory units; as expected given the ground truth architectures of those models.
>
> 5) *Clarifying the limitations of the model, training and dataset:*
>
> We highlight the limitations of the dataset in the main part of the manuscript; namely that we don’t train the ELM neuron in a biologically plausible way (e.g. similar to your suggestion), and that only the SHD and SHD-Adding are bio-inspired tasks, and the NeuronIO task is not. We highlight that the ELM neuron cannot give mechanistic insight into the functioning of synaptic plasticity of cortical neurons, and that other training setups are required to investigate the analogies in more detail.
>
> 6) *Future possible research directions:*
>
> We agree that exploring whether the ELM neuron would display similar I/O in the case of fitting it using bio-inspired techniques on bio-inspired tasks is an exciting avenue. However, we also agree that it is out of scope for the current manuscript. We plan to explore it in future research.

---

### Official Review · Reviewer_nAEZ · 2023-10-27

**Soundness:** 3 good
**Presentation:** 3 good
**Contribution:** 3 good
**Rating:** 8
**Confidence:** 3

**Summary:**

The authors introduced the Expressive Leaky Memory (ELM) neuron model, a bio-inspired model of a cortical neuron. It incorporates slowly decaying memory-like hidden states and a two-layered nonlinear integration of synaptic input. They showed that this model is able not only to capture the input-output mappings of cortical neurons efficiently but also to solve long-range dependency problems.

**Strengths:**

- The ELM model achieves a notable decrease in trainable parameters compared to temporal convolutional networks for simulating cortical neurons.
- The paper is well-written and easy to follow, particularly in the second section, where the design of the ELM neurons is explained.
- The selected experiments are suitable and effectively demonstrated the model's capabilities.

**Weaknesses:**

- The main text doesn't define the Branch-ELM variant. See questions.
- Minor training details need some clarifications. See questions.

**Questions:**

- A paragraph defining Branch-ELM would be necessary. Can you elaborate more on the intuition behind this variant? How does it work? When is it more suitable compared to the vanilla ELM? Why is it important to over-sample the input in this case?

- The term "fixed trainable" time constant is confusing. Is it a single tau value learned for each neuron that does not change after training?

- In Appendix B, the general training setup is detailed (batch size of 8, etc.). However, later on, different hyperparameters are used for the datasets. It is not clear where this general training setup was used.

- Does Figure S6 a) show any specific patterns?

---

> ### Author Response · Authors · 2023-11-20
>
> We thank the referee for evaluating our manuscript and for constructive criticism, in particular with respect to the need to clarify the Branch-ELM design and the used training parameters.
>
> 1) *A paragraph defining Branch-ELM would be necessary. Can you elaborate more on the intuition behind this variant? How does it work? When is it more suitable compared to the vanilla ELM? Why is it important to over-sample the input in this case?*
>
> We now have a dedicated description of the Branch-ELM at the beginning of the paper, together with the definition of ELM (in “The integration mechanism dynamics” paragraph), and generally expand on its discussion.
> The intuition: the Branch-ELM is inspired by the biological neuron, whose dendritic tree is composed of smaller hierarchically organized units called dendritic branches. While the dendritic tree overall might compute a highly nonlinear transformation, most inputs are summed linearly, and organization into branches guides where linear and where nonlinear summation is appropriate. We mimic this idea by slicing the input into a number (d_tree) of equally sized “branches” (for our best model, each synapse takes part in four slices, accounting for the mismatch between the ground truth and the branch assignment). Inputs are summed linearly within the slice. However, we let the downstream MLP take care of any further processing nonlinearities. If the slice size is large and the number of branches few, this approach reduces dramatically the number of parameters by reducing the size of MLP.
>
> - However, using this stark simplification, we lose general expressivity, and performance is expected to degrade unless the inputs are assigned to the synapse belonging to the correct ground truth branch identity and branch computations can be well approximated by a rectified sum.
>
> - As we only know that neighboring inputs mostly belong to the same branch identity, we sampled larger overlapping windows than the actual suspected branch sizes and let the model discard the incorrect assignments by modifying the synaptic weights.
>
> - In the appendix Figure S5 we show that randomly assigning synapses to branches or fixing the synaptic weights severely hurts performance, and having the incorrect branch activation like “sigmoid” or “relu” also impairs performance significantly.
>
> - In general, we expect the Branch-ELM to work great on datasets with large input dimensionality with low information density, such as SHD or SHD-Adding (coding for one or two digits per sample). We randomly oversample the input for the Branch-ELM on that dataset.
>
> 2) *The term "fixed trainable" time constant is confusing. Is it a single tau value learned for each neuron that does not change after training?*
>
> We agree that our phrasing was unfortunate, we meant “constant”, because they do not change during the execution, but they can be trained. We write now “constant explicit (trainable) timescales”.
>
> 3) *In Appendix B, the general training setup is detailed (batch size of 8, etc.). However, later on, different hyperparameters are used for the datasets. It is not clear where this general training setup was used.*
>
> Thank you for catching this inconsistency. There are parameters that can change across tasks (including the optimizer, batch size and initial learning rate). We report them now individually in the corresponding tasks.
>
> 4) *Does Figure S6a (now S8) a) show any specific patterns?*
>
> Overall the NeuronIO dataset contains mostly random biologically inspired firing patterns as input; this means varying ratios of inhibitory (blue) and excitatory (red) input, varying amount of input overall, varying degrees of input clustering, etc. The particular input pattern nicely shows this variation of excitation and inhibition over the course of two seconds. There is no structure beyond this in the input, unlike e.g. for the SHD or SHD-Adding dataset.

---

> > ### Comment · Reviewer_nAEZ · 2023-11-22
> >
> > Thank you for the additional information and the clarifications.

---

### Official Review · Reviewer_27bc · 2023-10-31

**Soundness:** 3 good
**Presentation:** 3 good
**Contribution:** 2 fair
**Rating:** 6
**Confidence:** 3

**Summary:**

This paper proposes artificial neural network models that incorporates important inductive bias (i.e. leaky memory dynamics, nonlinear synaptic integration) inspired from biological cortical neurons. The model is aimed to achieve two types of goals, one is to match the spike-level dynamics of pyramidal neuron, and the second is evaluated on bio-inspired tasks to evaluate temporal integration. For evaluation, they compare the model with others SOTA baselines (SNN, LSTM, Transformer) are evaluated on multiple biological inspired datasets and long sequence modeling task. It shows the benefits of efficiency in parameterization, and comparable or better performance compared to SOTA models. Meanwhile, the hyper-parameter tuning studies show overlap with previous literatures.

**Strengths:**

1. The paper is well-motivated, and take the reductionist view to minimize the parameters from a more detailed modeling for cortical neurons, and aim to address the computational efficiency needs of standard models.
2. Solid evaluations on multiple biological inspired datasets, and compared with multiple SOTA models, and follow by multiple hyperparameter tuning studies.
3. The biological realism side shows interesting overlaps with previous neuroscience literatures.
4. The results show the model is capable of achieving better accuracy with efficiency and fewer parameters than traditional deep model LSTM. The finding about simplification does not sacrifice the predictive performance is valuable.
5. The paper is well-written, and organized in a good structure.

**Weaknesses:**

1. This paper is aimed to balance the trade-off between fidelity, efficiency and biological realism. It did a fair job while still failed to capture some important aspects. For example, using MLP sacrifices the interpretability and biological realism to compensate accuracy. On the other hand, the model still sacrifices the accuracy and has a big performance gap when compared to SOTA models (S4 and Mega).
2. Scalability of the method: as scaling law plays a big role for improving transformer's predictivity, one concern is that transformer might perform better with increasing number of parameters. However, it might not be the same for ELM. As firstly shown in Fig 3, the accuracy quickly saturates with simply increasing $d_m$ and $d_{mlp}$, and not able to get further improved to minimize the performance gap between ELM and Mega.
3. Only one ML task is evaluated, more evaluations and benchmarks needed to demonstrate contributions in addressing long-range sequence modeling.
4. As shown in Table S1, large number of hyper-parameters still needed in advance or be tuned based on prior knowledge from neuroscience literatures.
5. The efficiency side might be over-claimed, as Table 1 shows ELM still requires 100k-200k parameters?

**Questions:**

1. What other critical components might be helpful to improve ELM model accuracy?
2. After applying sparse regularization or quantization to S4 and Mega to match the number of parameters to ELM, how much accuracy drop they will have?
3. Is the model able to be trained with other biological plausible learning rules instead of BPTT? How they might end up with different parameters?

---

> ### Author Response · Authors · 2023-11-20
>
> We thank the referee for evaluating our manuscript and for constructive criticism, in particular with respect to expanding on the scalability of the ELM neuron.
>
> 1) *Clarification of MLP design choice of ELM neuron:*
>
> We will include in the manuscript that one of the key modeling goals of the ELM design was to quantify the degree of nonlinearity necessary to model a neuron’s dynamics; hence the choice of MLP as it offers a straightforward way to do this using $d_{mlp}$ and $l_{mlp}$. In contrast, in more biologically realistic detailed biophysical models, there is no straightforward way to quantify the amount of nonlinearity in such models, and therefore offers less interpretability along this dimension compared to the ELM neuron.
>
> 2) *Clarification of saturating performance on NeuronIO:*
>
> We will add to the NeuronIO task description that the underlying Biophysical model displays chaotic dynamics, meaning any surrogate model's performance will satire below perfect prediction (see also LSTM and TCN), due to the inherent unpredictability of the task. We additionally provide LIF and ALIF models *(see additional pdf)* as an example of fitting neuron models that do not display chaotic dynamics; the ELM neuron performance saturates close to perfect predictability in this case.
>
> 3) *Clarification of tuning ELM parameters:*
>
> We will highlight that the ELM neuron primarily only requires tuning along the memory $d_m$ dimension (and $\tau_s$ for very sparse tasks). We add the reference to the text for the hyper-parameters tuning table in the appendix that highlights how other hyperparameters can be derived from $d_m$, the input dimensionality $d_s$, and the task length $T$, and what default parameters should be used otherwise. A further testament to this is the LRA task, where mostly the same hyper-parameters across all tasks. Other training parameters such as batch size, learning rate, etc. still need tuning of course. Therefore, we believe the ELM neuron does not require more tuning than e.g. the LSTM.
>
> 4) *Number of Long-range ML tasks:*
>
> There might be a misunderstanding concerning the number of tasks. The LRA benchmark contains 6 rather different tasks. Additionally, the Spoken Heidelberg Digits Task (SHD), and particularly SHD-addition, has a context length of 2000 for a bin size of 1, and therefore is twice as long as, e.g., the Image or Pathfinder task from LRA. Therefore, we provide results on a total of 8 long-range tasks.
>
> 5) *Addressing when the ELM neuron can be expected to be efficient:*
>
> We agree that this needs further clarification. The ELM and in particular Branch-ELM neuron can mainly be expected to be much more efficient if the input dimensionality is very large, such as in the NeuronIO task ($d_s=1278$), or somewhat in the SHD and SHD-Adding task ($d_s=700$), and less so e.g. on LRA ($d_s = 8$ to $d_s = 169)$; also note the increased relative performance for language tasks with higher input dimensionality. We now elaborate this condition for efficiency more clearly in the manuscript.
>
> 6) *Addressing the scalability and limitations of the ELM neuron:*
>
> We acknowledge the scaling limitation of individual ELM neurons, which we set out to explore using the LRA benchmark, however, we would like to point out that the training was kept fairly simple, and we did not use e.g. warmup, weight decay, layer-norm, dropout, etc (which the comparison models did). Additionally, we suspect that assembling such complex neurons in more sophisticated multi-layered networks might provide the desired scalability; unfortunately, this is beyond the scope of this work. Nevertheless, we provide an ablation of the ELM neuron and  S5 model on Pathfinder in terms of parameters in Tables S5 and S6 showing that an ELM with 3.5K parameters is outperforming a single layered S5 with ~170K parameters (62.95% ACC vs 58.19% ACC). Furthermore, motivated by the observation that SOTA models on LRA use various regularization techniques, we train ELM neuron with various dropout probabilities on Pathfinder in Table S7, and find that models trained with 0.3 dropout probability for 1000 Epochs can reach ~82\% accuracy, and improvement of 10% accuracy, evidencing that the performance gap might be reduced through additional proper regularization techniques.
>
> 7. *Training with biological learning rules:*
>
> We agree that it would be interesting to study the biologically plausible training, and it might happen that the training will converge to a different solution. However, for this manuscript, we aimed to establish how the model can perform with standard machine learning optimizers (that typically define the upper limit on the performance, seldomly achieved by the biologically plausible rules). Finding how to train this model with, for example, Eq.Prop or even more complicated bio-plausible plasticity rules would be an exciting direction. We added it to future work suggestions.

---

> > ### Comment · Reviewer_27bc · 2023-11-23
> >
> > Thanks for the authors' clarifications and efforts to add clarifications and new experiment results (saturation on prediction of chaotic dynamics; number of ML tasks; scalability, improvement by regularizations). I have updated my score correspondingly.

---

### Official Review · Reviewer_YDMm · 2023-11-01

**Soundness:** 3 good
**Presentation:** 1 poor
**Contribution:** 4 excellent
**Rating:** 8
**Confidence:** 4

**Summary:**

The authors propose a phenomenological neuron model called the Expressive Leaky Memory (ELM) neuron that uses various biologically inspired features. Specifically, it has separate synapse and memory dynamics and an integration mechanism defined by a learnt MLP. The model also allows learning of the various time constants. The authors demonstrate that this model is able to fit the input-output relationship in a dataset generated from a detailed biophysical model. Moreover, the authors show that this model can perform long-range dependency modelling better than vanilla transformers and a LSTM-based recurrent model.

**Strengths:**

- The paper is well motivated, and the model uses abstract simplifications to represent known biological details. This provides a relatively parsimonious but abstract model to model biological neurons, which is a very interesting approach and novel to my knowledge.
- The fact that this model, even though abstract and motivated by biology, still performs well in long range arena is very interesting.
- The description of related work is very comprehensive.
- The authors include a good discussion of the potential shortcomings of the model.
- Overall, the quality and significance of this work is high.

**Weaknesses:**

- There are major clarity issues in the paper. Many aspects of the model and notation are unexplained (e.g. $\lambda$, $1-\kappa_m$ in Fig. 1(c)). The explanation of Branch-ELM comes much later, even though it's referred to multiple times before that, which makes it very hard to read. The role of $w_s$ is also not clear at all (the given explanation on Pg. 3 doesn't help).
- The behaviour of Branch-ELM is unclear -- if the input is shuffled, does it affect performance? Since it depends on the local window to group inputs?
- It's a bit odd that ELM doesn't perform well for short sequences (large bin size) as seen in Fig. 5 whereas LSTM does. The performance of ELM for short sequences could be explored more, since it sounds like that might be a major shortcoming of the model.
- It is not clear how the number of parameters for the various cases were chosen.
- I think exploring the multi-layer case would have made the paper much stronger. It's also not clear if this was avoided because of the computational constraints, since for mid-size LSTMs at least, multi-layer networks still fall very much in the computationally tractable regime.

**Questions:**

## Questions

- Would this neuron be able to model synapse dynamics such as short-term plasticity (Tsodyks et al. 1998)?

## Suggestions

- Spike frequency adaptation for spiking neurons was proposed in (Bellec et al. 2018) rather than (Bellec et al. 2020).

### Minor:

- In the abstract, "exploiting a few such slowly decaying..." sentence reads very odd.
- sentence above beginning of Sec. 4: "puting the major emphesis" has typos. That paragraph is very hard to read.

(Bellec et al. 2018) Bellec, G., Salaj, D., Subramoney, A., Legenstein, R., and Maass, W. (2018). Long short-term memory and Learning-to-learn in networks of spiking neurons. In Advances in Neural Information Processing Systems 31, pp. 787–797.

(Tsodyks et al. 1998) Tsodyks M, Pawelzik K, Markram H. Neural networks with dynamic synapses. Neural Comput 10: 821– 835, 1998.

---

> ### Author Response · Authors · 2023-11-20
>
> We thank the referee for evaluating our manuscript and for constructive criticism, in particular concerning how the manuscript's presentation can be improved for clarity.
>
> 1)  *Better introduction and explanation of the Branch-ELM neuron model:*
>
> We improve the presentation of our paper by introducing the Branch-ELM together with the ELM neuron in section 2, by additionally expanding on the parameters involved in scaling the MLP output, adding references to the supplementary tables and by clarifying the Branch-ELM processing behavior in detail.
>
> 2) *Clarification of the parameters scaling of the MLP output:*
>
> The constant $\lambda$ (float constant) and $1-\kappa_m$ (float vector, dependent on $tau_m$) are factors that scale the output of the MLP. Together, they ensure that the memory values $m_t$ are constrained to stay within a range of [-$\lambda$, $\lambda$] if $tanh$ range is [-1,1], which prevents exploding gradients.
> We now added a short explanation to the formulation of the model (in “The integration mechanism dynamics” section)
>
> 3) *Clarification of the Branch-ELM processing of its inputs on NeuronIO:*
>
> Indeed, this is correct; if the ordering of input channels is shuffled, performance degrades significantly for the Branch-ELM. Neighboring input channels are typically located next to each other in the biophysical model dataset we are using, hence we oversample the input in a moving window fashion to exploit this insight (the resulting overlap between branches is to account for uncertainty in exact ground truth branch bounds). We tested the shuffling effect in Supplementary Figure S5 (purple trace), where random synapse-to-branch assignment makes the model incapable of reaching the spike-prediction threshold. For other datasets, such as SHD, however, we randomly assign synapses to branches. We will highlight this in the main text and refer to additional branch-related ablations in the appendix.
>
> 4) *Clarification of ELM neuron and LSTM performance for larger bin sizes on SHD:*
>
> The ELM and Branch-ELM neurons consistently outperform the LSTM for all binning sizes. Nevertheless, performance degrades for larger bin sizes for all models, as the input binning is a lossy pre-processing step, where temporal information is eliminated. Performance improves with the bin size for LSTM because, for the small time bins, LSTM could not take advantage of the whole temporal structure but gets into training problems due to the many time bins it has to keep in memory. The information is lost as the bin size increases, but the small number of bins enables LSTM to learn the task. We show that the temporal structure of the task is not a problem for ELM neurons, so they are only facing the drawbacks of the coarser data discretization. Overall, the observed performance dynamics is not a shortcoming of, e.g. the ELM or LSTM models but a consequence of the data preprocessing. We now discuss it in the main text, and include a statement about the hyper-tuning on the dataset in the appendix.
>
> 5) *Correspondence between stacked ELM neuron and biological neuron unclear:*
>
> We agree that it is a promising direction of future investigation, particularly with the focus on increasing performance. Our main focus for the current contribution is to explore the performance and limitations of the single expressive neuron. To maintain biological correspondence, it would be interesting to train the recurrent network of ELM neurons (mimicking the recurrent nature of computations in the brain).
>
> 6) *Comparison to matched parameter performance on LRA between:*
>
> To provide some evaluation results with matched parameters on LRA, we provide an additional ablation on Pathfinder of the number of memory units for the ELM neuron and number of layers for the S5 model respectively; now displayed in Tables S5 and S6 The results show that an ELM with 3.5K parameters is outperforming a single layered S5 with ~170K parameters (62.95% ACC vs 58.19% ACC).
>
> 7) “Would this neuron be able to model synapse dynamics such as short-term plasticity (Tsodyks et al. 1998)?”
>
> At the moment, the ELM neuron does not have a dedicated mechanism for short-term plasticity. One reason for this is that STD is primarily implemented on the pre-synaptic side (using depletion of synaptic resources). However, we suspect that, in principle, the ELM neuron can capture various short-term plasticity effects dependent solely on internal processes by leveraging a sufficient number of memory units (much larger than used now). We tested the performance of ELM on the simplified neuron with spike-frequency adaptation (aLIF) that is somewhat similar to the STD effect but is completely controlled by the neuron (see Appendix Table S4).

---

> > ### Comment · Reviewer_YDMm · 2023-11-22
> >
> > Thank you for the clarifications and updates. I've updated my score to reflect the addressing of the clarity issues.
> >
> > Minor note: The spike-frequency adaptation reference is still pointing to the wrong paper in the revised version.

---

> > > ### Author Response · Authors · 2023-11-22
> > >
> > > Thank you very much for your consideration! We apologize for the oversight; and have updated the incorrect refference.

---

### Author Response · Authors · 2023-11-20

We would like to thank all the reviewers and the area chair for their valuable comments and considerations. We contributed multiple new experiments and made several changes in the manuscript to improve its quality significantly.

Furthermore, we are grateful to the reviewers for recognizing our various contributions; in particular, the ELM model’s ability to efficiently represent a pyramidal neuron’s I/O was highlighted (YDMm, 27bc, nAEZ, 8X2o). Reviewers noted the paper's solid motivation (YDMm, 27bc, 8X2o), its well-written nature (27bc, nAEZ, 8X2o), and commended the comprehensive related work and the thorough discussion of its limitation (YDMm). The evaluations were deemed solid (27bc),  and suitable to demonstrate the model's capabilities (nAEZ). The model's ability to recover biological detail (27bc), novel approach to abstracting biological neurons (YDMm),  and ability to learn long range dependencies (YDMm, nAEZ), was recognized.

In response to the remaining critiques and questions, we made the following improvements:

1) Additional performance ablations of ELM on pathfinder

=> showing possibility to further improve performance on LRA benchmark

2) Additional model size ablations of ELM & S5 on pathfinder

=> giving better insight into ELM model efficiency on LRA benchmark

3) New experiments fitting ELM to LIF & ALIF phenomenological neuron models

=> further evidencing ELM as suitable model to get insight on underlying neural computations

4) New experiments fitting LIF & ALIF to the NeuronIO dataset

=> highlighting the utility of the ELM neuron compared to other phenomenological models

5) Snooner introduction and Improved description of Branch-ELM workings

6) More precise formulation of contributions and the ELM neuron capabilities

---

### Meta-Review · Area_Chair_J8K5 · 2023-12-11

**Metareview:**

The paper introduces the Expressive Leaky Memory (ELM) neuron model, a biologically inspired model for emulating cortical neuron dynamics. It demonstrates the model's efficiency in processing long-range temporal dependencies, performing well against traditional architectures like Transformers and LSTMs.

Strengths: balancing biological realism with computational efficiency.

Weaknesses: clarity issues; questions about scalability

**Justification For Why Not Higher Score:**

The paper is recommended for poster presentation due to outstanding concerns about scalability and the need for deeper exploration of the model's limitations.

**Justification For Why Not Lower Score:**

The paper is above the acceptance threshold.

---

### Decision · Program_Chairs · 2024-01-16

Accept (poster)